

# Light absorption by polar and non-polar aerosol compounds from laboratory biomass combustion

Deep Sengupta,[1] Vera Samburova,[1] Chiranjivi Bhattarai,[1] Elena Kirillova,[2] Lynn Mazzoleni,[2] Michealene Iaukea-Lum,[1] Adam Watts,[1] Hans Moosmüller,[1] Andrey Khlystov[1]

[1] Desert Research Institute, 2215 Raggio Parkway, Reno, NV 89512, USA
[2] Michigan Technological University, 1400 Townsend Drive, Houghton, MI 49931, USA

*Correspondence to*: vera.samburova@dri.edu

**Abstract.** Fresh and atmospherically aged biomass-burning (BB) aerosol mass is mostly comprised of black carbon (BC) and organic carbon (OC) with its light-absorbing fraction – brown carbon (BrC). There is a lack of data on the physical and chemical properties of atmospheric BB aerosols, leading to high uncertainties in estimates of the BB impact on air quality and climate, especially for BrC. The polarity of chemical compounds influences their fate in the atmosphere including wet/dry deposition and chemical and physical processing. So far, most of the attention has been given to the water-soluble (polar) fraction of BrC, while the non-polar BrC fraction has been largely ignored. In the present study, the light absorption properties of polar and non-polar fractions of fresh and aged BB emissions were examined to estimate the contribution of different-polarity organic compounds to the light absorption properties of BB aerosols.

In our experiments, four globally and regionally important fuels were burned under flaming and smoldering conditions in DRI's combustion chamber. To mimic atmospheric oxidation processes (5-7 days), BB emissions were aged using an oxidation flow reactor (OFR). Fresh and OFR-aged BB aerosols were collected on filters and extracted with water and hexane to study absorption properties of polar and non-polar organic species. Spectrophotometric measurements over the 190 to 900 nm wavelength range showed that the non-polar (hexane-soluble) fraction is 2-3 times more absorbing than the polar (water-soluble) fraction. However, an increased absorbance was observed for the water extracts of oxidized/aged emissions while the absorption of the hexane extracts was lower for the aged emissions. Comparing the absorption Ångström Exponent (AAE) values, we observed changes in the light absorption properties of BB aerosols with aging that was dependent on the fuel types. The light




absorption by HUmic LIke Substances (HULIS) was found to be higher in fuels characteristic of the southwestern USA. The absorption of the HULIS fraction was lower for OFR-aged BB emissions. Comparison of the light absorption properties of different polarity extracts (water, hexane, HULIS) provides insight into the chemical nature of BB BrC and its transformation during oxidation processes.

**Keywords**. Biomass burning, organic aerosols, brown carbon, light absorption, non-polar organic fraction, oxidation flow reactor (Potential Aerosol Mass reactor)



# 1 Introduction

Biomass burning (BB), including wildfires and controlled burns, can contribute significantly to the atmospheric aerosol loading (Park et al., 2007). BB emissions alter regional air quality (Liu et al., 2009), cause radiative forcing and contribute to climate change (Jacobson, 2004), reduce visibility (Lee

et al., 2016), and cause adverse health effects (Arbex et al., 2007; Regalado et al., 2006). Biomass combustion is a major source of greenhouse gases (e.g., $CO_2$), volatile organic compounds (VOCs), $NO_x$, CO, and carbonaceous aerosols in the atmosphere (Finlayson-Pitts, B. J. and J. N. Pitts, 1999). In addition to greenhouse gases, carbonaceous particles can perturb the radiation budget of the Earth's atmosphere (Hobbs et al., 1997; Podgorny et al., 2003) due to their light scattering and absorbing

properties (Corr et al., 2012; Liu et al., 2016a).

Just a decade ago, black carbon (BC) was considered to be the only light absorbing compound in atmospheric carbonaceous aerosols (Ramanathan et al., 2001). Based on this premise, climate model simulations used known BC mass concentrations to compute aerosol absorption optical depths, which were compared with surface and satellite-based observations (Holben et al., 1998; Torres et al., 2007).

Discrepancies between simulated radiative forcing estimates with known BC mass concentrations and observed absorption values were high in regions dominated by BB aerosols (Koch et al., 2009). Both laboratory-based studies (Shiraiwa et al., 2010) and field campaigns (Cappa et al., 2012; China et al., 2013; Gyawali et al., 2017; Lack and Cappa, 2010) offer compelling evidence in favor of the ubiquitous existence of non-BC light-absorbing compounds in carbonaceous aerosols. The light absorbing organic

compounds are known as brown carbon (BrC), mostly due to their brownish color (Formenti, 2003). They are abundant in BB aerosols and strongly absorb light in the UV and near-UV visible region of the spectrum (Bond and Bergstrom, 2006; Kirchstetter et al., 2004; Moosmüller et al., 2009).

BrC chemical composition and light absorption efficiency vary depending on the fuels, burning conditions (Akagi et al., 2011; Jaffe and Wigder, 2012), and degree of atmospheric oxidation (Andreae

and Gelencsér, 2006; Hoffer et al., 2006). In order to understand the global impact of BB organic aerosols (BBOA) on atmospheric processes and climate, comprehensive characterization of their BrC fraction generated during combustion of relevant fuels from different geographical locations is needed.



BB processes and the subsequent atmospheric aging (Andreae, 1997) can produce a wide range of compounds including water-soluble HULIS (Graber and Rudich, 2006) and polycyclic aromatic hydrocarbons (PAHs) (Samburova et al., 2016). However, very little is known about light absorption properties of other BB organic compounds.

In the present study, we investigated BB emissions from several globally and regionally important fuels and characterized light absorption by BrC in BBOA extracts of different polarity. While the water-soluble fraction of BB aerosols has attracted much attention (Kiss et al., 2002; Mayol-Bracero et al., 2002), only a few studies (Chen and Bond, 2010) have reported light absorption properties of non-polar BBOA extracted with lower-polarity solvents. Characterization of light absorbing compounds extracted

with different polarity solvents can greatly improve our understanding of the BB aerosol chemistry, wet and dry deposition of BB constituencies, and effects of BB aerosols, in particular of their BrC fraction, on regional and global radiative forcing. Spectrophotometric characterization of different polarity BB extracts was performed to analyze their light absorption properties in both fresh and oxidized states. These results were supplemented with ultrahigh resolution Orbitrap Elite mass spectrometry

observations of the high molecular weight fraction of the water-soluble aerosol.

## 2. Experiments

### 2.1 Fuel Description

Several globally and regionally important BB fuels were selected for the present study; they can be divided into two categories: (1) fuels that burn mostly with smoldering combustion and (2) fuels that

burn mostly with flaming combustion.

**Smoldering Combustion:** Florida peat from southeastern USA, used for our study, is a semi-tropical fuel and consists of organic muck and swamp soils (Watts, 2013; Watts and Kobziar, 2013). Wetlands in the southeastern USA are currently under tremendous stress due to urbanization and increasing demand for water resources. The resulting desiccated wetlands are considered to be fire prone zones and

can cause large carbon emissions during wildland fires.

Eurasian peatlands host a large part of the global carbon pool (Yu, 2012) and are also highly vulnerable to wildfires due to climatic warming, permafrost degradation, and prolonged fire seasons (Brown et al.,




2015; Turetsky et al., 2015). For this study peat samples were collected from the Pskov region of Russia. The indigenous species that are responsible for peat formation in this region are Sphagnum and cotton grass (Eriphorum spp.).

**Flaming Combustion:** For flaming combustion fuel, we collected vegetation from the Fishlake

National Forest, UT, USA where the Joint Fire Science Program (JFSP) of the U.S. land management agencies is planning to conduct large prescribed burns as part of the Fire and Smoke Model Evaluation Experiment (FASMEE). Dominant species are subalpine fir (*Abies lasiocarpa*) and quaking aspen (*Populus tremuloides*); minor species include Douglas fir (*Pseudotsuga menziesii*), limber pine (*Pinus flexilis*), and common juniper (*Juniperus communis*). Another flaming combustion type fuel was

collected in the immediate vicinity of a local, Reno, NV, fire event that occurred in June 2016 (Hawken Fire) on the lower eastern slope of the Sierra Nevada's Carson Range. This fuel is characteristic of vegetation of the southwestern U.S. and is dominated by grasses, shrubs, shadscale (*Atriplex confertifolia*), sagebrush (*Artemisia arbuscula*), and antelope brush (*Purshia tridentata*).

**2.2 Reagents and Materials**

PAH standards were acquired from Sigma-Aldrich (St. Louis, MO, USA), AccuStandard (New Haven, CT, USA), and Cambridge Isotope Laboratories, Inc. (Andover, MA, USA). High-performance liquid chromatography (HPLC) grade methanol and hexane were obtained from Fisher Scientific (Fair Lawn, NJ, USA). Nano-Pure water was generated with a Barnstead Nanopure instrument (Thermo Scientific,

Dubuque, IA, USA). Solid phase extraction was performed using Waters Oasis HLB cartridges (Waters, Milford, MA, USA). Three different kinds of filters were used for sampling and further chemical analyses: 1) pre-fired 47-mm diameter quartz-fiber filters (2500 Pallflex QAT-UP, Pall Life Sciences, Ann Abor, MI, USA) for thermo-optical EC/OC analysis, 2) Teflon filters (2500 Pallflex QAT-UP, Pall Life Science, Ann Abor, MI, USA) for gravimetric particulate matter (PM) mass analysis, and 3)

Teflon-impregnated glass fiber (TIGF) 47-mm diameter filters (Fiber FilmT60A20, Pall Life Sciences, Ann Abor, MI, USA) for organic analysis.



### 2.3 Biomass-Burning Experiments

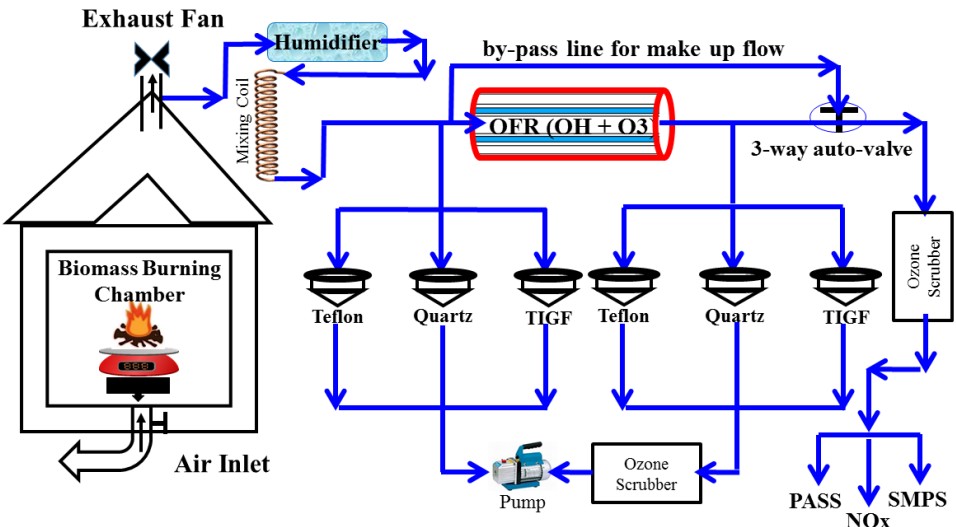

**Figure 1.** DRI Biomass Burning (BB) Facility with Oxidative Flow Reactor (OFR) and flow setup.

BB experiments were conducted using DRI's BB facility for combustion of the selected fuels under controlled conditions. A close replicate of this facility was described previously (Tian et al., 2015) and a detailed discussion of the experimental set up can be found elsewhere (Bhattarai et al., 2017). Fuels were placed on a burn platform that consists of a ceramic disk ($d$ = 45.7 cm) on top of a balance (Veritas L Series Precision Balance, Hogentogler & Co. Inc., Columbia, MD, USA, 0.01 g precision). A

propane burner (Worthington Cylinder Cooperation Columbus, OH, USA) was used to initiate the burns. During each burn, the mass on the burn platform was recorded every 2 seconds. A burn was considered complete when no significant change in fuel mass was observed for two and more minutes.

Laboratory generated BB emissions were mixed with humidified zero air (Airgas Inc., Sparks, NV, USA) using a 4-m long spiral copper tubing (12.7 mm OD). Before the mixing with BB emissions, the

zero air was humidified by bubbling through Nano-pure water in a glass 500-mL volume impinger. The flow rate was controlled by a mass flow controller (810C-CE-RFQ-1821, Sierra Instruments, Monterey,



CA, USA). A Potential Aerosol Mass (PAM) Oxidative Flow Reactor (OFR) (Aerodyne Research Inc., Billerica, MA, USA) was used to mimic approximately seven days of atmospheric aging. The OFR consists of an alodine-coated aluminum cylinder (46 cm length and 22 cm diameter) with an internal volume of 13.3 L. Two sets of UV lamps emit UV-radiation at wavelengths of 185 and 254 nm

(Atlantic Ultraviolet Corporation, Hauppauge, NY, USA; part number GPH436T5VH/4P & GPH436T5L/4P) in the OFR to produce ozone and OH radicals (Li et al., 2015). UV irradiance in the OFR was quantified with a photodiode detector with wavelength range of 225-287 nm. [TOCON_C6; Sglux GmbH, Germany].  Ultra-high purity nitrogen (Airgas Inc., Reno, NV, USA) was used to purge the UV lamp compartments to prevent the lamps from overheating. A probe that monitors relative

humidity and temperature inside the OFR (from Aerodyne Inc., MA, USA) was mounted towards the outlet side of the OFR.

Fresh (directly from the chamber) and aged (oxidized in the OFR) emissions were continuously collected on separate sets of Teflon and pre-fired quartz filters, as well as on a TIGF filter followed by an XAD cartridge for detailed chemical analysis. Several online instruments were used to characterize

gas and particle phase pollutants (see Figure 1). The online instruments alternated every 10 min between sampling fresh and aged emissions using a computer-controlled three-way valve system. In this study, data collected with a chemiluminescence NOx analyzer (Thermo-Environmental Instruments Inc., Franklin, MA 02038, USA) were synchronized with filter measurements. $NO_X$ concentrations in fresh and aged BB emissions were measured consecutively. $NO_2$ concentrations for individual burns

were normalized by consumed fuel mass ($\Delta m$) yielding fuel-based emission factors for fresh and aged emissions separately. These $[NO_2]/\Delta m$ emission factors for fresh and aged emissions are subsequently averaged over the total run time for each burning session.

A bypass flow was used to keep the flow from the BB chamber and through the OFR constant when online instruments switch between sampling fresh and aged emissions.  To protect online instruments

from high ozone produced in the OFR, ozone scrubbers were installed in front of the instruments' inlets. The ozone scrubbers were loaded with charcoal followed by Carulite 200 catalyst (Carus Corp., Peru, IL, USA).

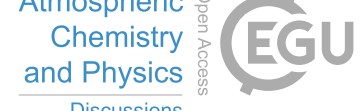

All burning experiments, except for Siberian peat combustion, were conducted with 2.1 V of OFR voltage, which is equivalent to approximately seven days of aging (Bhattarai et al., 2018). Since no replicates were collected during Siberian peat combustion, standard deviations and mean values were not calculated for these burns. Instead, during the combustion of Siberian peat, three different OFR

voltages (3, 5, and 7 V) were used to monitor the effect of enhanced oxidation with increasing OFR voltage and thus oxidants concentration.

### 2.4 Extraction Procedure

Teflon filters were extracted for 20 min under sonication in 30 mL of Nano-Pure water. The extracts

were filtered with PTFE filters (0.22 μm pore size, ThermoScientific, TN, USA); 20 ml of the extract was kept for further chemical analysis. 2 mL aliquots of each extract were used for spectrophotometric analysis and 8 mL were used for the solid phase extraction (SPE) and further HULIS analysis. Prior to the SPE procedure, the pH of the aqueous aerosol extract was adjusted to 2 with 1 M HCl (Varga et al., 2001). This ensures retention of compounds with multiple polar functional groups (HULIS) in the SPE

column (Kiss et al., 2002; Varga et al., 2001). At pH 2, all functional groups are expected to be protonated and less hydrophilic than at neutral pH ($\sim$ 6-7). To remove the residue of inorganic constituents, SPE cartridges were rinsed with water and dried for 30 min. The retained organics were eluted with 6 mL of methanol.

Hexane extracts were prepared using one half of each of the quartz filter samples. Prior to hexane

extraction, the filter halves were spiked with deuterated PAH standards needed for future chemical analysis of these extracts. The spiked quartz filters were extracted using an accelerated solvent extractor (ASE) (DIONEX, ASE-300, Salt Lake City, UT, USA). All hexane extracts were concentrated on a rotary evaporator (Rotavapor R-124, BÜCHI, New Castle, DE, USA) to 6 mL. 2 mL of the concentrated hexane extract was used for spectrophotometric analysis (Lambda 1050, PerkinElmer, Waltham, MA,

USA).



### 2.5 OC/EC Analysis

Samples collected on quartz-fiber filters were used to estimate total organic carbon ($OC_{Total}$) and elemental carbon (EC) mass. Punches (area = 1.5 cm$^2$) from the quartz filter samples were analyzed with a thermal-optical carbon analyzer (Atmoslytic Inc., Calabasas, CA, USA) following the

IMPROVE protocol (Chow et al., 1993, 2004). To obtain carbon content of hexane and water extracts separately, the approach of Lowenthal et al. (2009) was adopted. 30 µl hexane extract aliquots were transferred to 7-mm diameter pre-baked quartz filter punches. The filter punches were dried for 24 hours allowing the hexane to evaporate completely. The punches were then analyzed using the thermal/optical carbon analyzer to obtain the total OC content of the hexane extracts ($OC_{Hexane}$). To

correct for the background, five blanks were prepared by spiking pre-baked filter punches with pure hexane and analyzed in the same way. Mean OC values of the blanks (0.7 µg/punch) were subtracted from total $OC_{Hexane}$ values (on average 2.4 µg/punch for all fuels). The OC mass of the hexane extract $OC_{Hexane}$ (in g ml$^{-1}$) was subtracted from the total OC mass ($OC_{Total}$) yielding the OC mass of the polar OC fraction ($OC_{Water}$) under the assumption that compounds that are not soluble in hexane, are water

soluble.

### 2.6 Spectrophotometer Analysis

Spectrophotometric characterization of water, hexane, and water extracts after SPE (HULIS fraction in methanol) were performed with a UV–Vis spectrophotometer (Lambda 1050, PerkinElmer, Waltham,

MA, USA) in 10-mm path length quartz macro cuvettes (FireflySci. Inc., NY, USA). Absorption spectra were taken over the 190 – 900 nm wavelength range. The spectrophotometer was operated with a reference cell, filled with the corresponding blanks: pure water for water extracts, hexane with PAH standards for hexane extracts, and methanol - for SPE extracts. The use of corresponding blanks in the reference cell helped to eliminate/subtract absorption caused by solvents and internal standards. Results

are reported in terms of absorbance $A$, defined as $A = \log I_0/I$ (where $I_0$ is the radiant power of incident light and $I$ is the radiant power of transmitted light) and related to the transmittance $T$ by $A = -\log_{10} T$, to the optical depth $\tau$ by $A = \tau/\ln 10$. The relation between absorbance at each wavelength $A_\lambda$ and the





fractional volume concentration of the analytes ($c$) is given by Eq: 1a where $\epsilon$ is absorption coefficient and $l$ is length of cuvette (1cm).

$$A_\lambda = \epsilon c l \tag{1a}$$

Extinction coefficient can be calculated rearranging Eq (1a) as

$$\epsilon = \frac{A_\lambda}{cl} . \tag{1b}$$

The baseline correction followed the approach of Hecobian et al., (2010) subtracts the mean absorbance $A_{Mean(700-900)}$ in the wavelength range of 700 to 900 nm and  from the absorbance values for each individual wavelength $A_\lambda$ to yield the baseline-corrected absorbance $AbS_\lambda$ as

$$AbS_\lambda = (A_\lambda - A_{Mean(700-900)}) \tag{1c}$$

10    It is important to consider how the absorbance values are contributing to solar radiative forcing estimates. Clear sky total spectral solar irradiance on a tilted receiver plane ($GTIR_\lambda$) was determined with the Simple Model of Atmospheric Radiative Transfer of Sunshine (SMARTS) program (Gueymard, 1995). The normalized spectral irradiance of the solar radiation was multiplied with obtained absorbance spectra for all extracts to estimate solar weighted absorbance ($[AbS_\lambda]_{SW}$).

$$[AbS_\lambda]_{SW} = AbS_\lambda \times \frac{GTIR_\lambda}{\sum_{290}^{900} GTIR_\lambda} \tag{2}$$

The total absorbance ($TotalAbS$) was calculated by integrating solar weighted absorbance from 190 to 900 nm and normalizing it with respect to the total consumed fuel mass during each burning experiment. To compare the total absorbance between different extracts, all dilution factors were taken into account.

$$TotalAbS = \frac{\sum_{290}^{900} [AbS_\lambda]_{SW}}{fuel\ mass\ consumed} \tag{3}$$

We performed two tailed pairwise t-tests on sets of fresh and photo-chemically aged (with OFR) aerosols in different extracts to evaluate the statistical implications of our results. We computed the t-statistic under the null hypothesis of "no change in absorption took place upon aging/oxidation". All p-values from the statistic are reported in Table 2 (Supplementary section).





In order to estimate the absorption efficiency of the extracts we applied the same strategy like (Eq. 1b) and calculated the mass absorption coefficients (MAC) were calculated for both hexane and water extracts (length of the spectrophotometer cuvette is 1 cm).

$$\text{MAC}_{\text{Hexane}} = \{[AbS_\lambda]_{SW}\}_{\text{Hexane}} / \text{OC}_{\text{Hexane}} \tag{4}$$

$$\text{MAC}_{\text{Water}} = \{[AbS_\lambda]_{SW}\}_{\text{Water}} / \text{OC}_{\text{Water}} \tag{5}$$

The solar weighted MAC values (MAC$_{sw}$) from each individual wavelength were integrated over the entire spectral range yielding Spectrally Integrated Solar weighted Mass Absorption Efficiency or SIMAC$_{sw}$.

Absorption spectra often can be approximated by a power law with a single exponent, the so-called
Absorption Ångström Exponent (AAE), quantifying the wavelength dependence of the absorbance as

$$\frac{AbS_{\lambda 2}}{AbS_{\lambda 1}} = \left[\frac{\lambda 1}{\lambda 2}\right]^{AAE}, \tag{6}$$

where $AbS_\lambda$ is the absorbance at the corresponding wavelength $\lambda$. To calculate AAE, Eq. (6) can be written as

$$AAE = -\frac{ln(AbS_{\lambda 2}) - ln(AbS_{\lambda 1})}{ln(\lambda 2) - ln(\lambda 1)}, \tag{7}$$

where one can clearly see that AAE is the negative slope of absorbance plotted as function of wavelength in log-log space (Moosmüller et al., 2011). Here, we use the AAE values for three different wavelength ranges 1) 200 – 400 nm, 2) 400 – 600 nm, 3) 600 – 900 nm, each derived with linear regression in log-log space. Absorbance data in the 600 – 900 nm range were quite noisy; therefore, we averaged sets of ten data points, thereby reducing the number of data points from 300 to 30 and
obtaining AAE from a linear regression of these points in log-log space. We determined AAE values for extracts of fresh and aged aerosol emission from the combustion of several different fuels.

The imaginary part of the bulk refractive index was calculated for water extracts. MAC values computed from spectrophotometer analysis were multiplied by mass density ($\rho$) of the aerosols (see Eq.(8)) to obtain absorption coefficients ($\beta$) such as used in Mie theory (Sun et al., 2007).

$$\beta = \rho MAC_{water}. \tag{8}$$





The mass density of bulk aerosol depends on fuel type and burning condition. In this work, instead of making an attempt to measure aerosol density for all fuel types (and both fresh and aged aerosols), we assumed a general BrC aerosol mass density of 1569 kg m$^{-3}$ (Hoffer et al., 2006) for all BB aerosols

from our experiments. The bulk absorption coefficient $\beta$ is related to the imaginary part of the refractive index ($k$) (Moosmüller et al., 2009) as

$$\beta = \frac{4\pi k}{\lambda} \tag{9}$$

and the imaginary part of the refractive index $k$ can be written as

$$k = \frac{\beta\lambda}{4\pi} . \tag{10}$$

Total imaginary refractive index $k_{total}$ was computed by using a simple volume mixing rule as

$$Hexane\ Fraction(HF) = \frac{OC_{Hexane}}{OC_{Hexane}+OC_{Water}} \tag{11.a}$$

$$Water\ Fraction(WF) = \frac{OC_{water}}{OC_{Hexane}+OC_{Water}} \tag{11.b}$$

$$k_{total} = HF*k_{Hexane} + WF*k_{water} , \tag{11.c}$$

under the assumption that OC$_{Hexane}$ and OC$_{Water}$ have the same mass density

### 2.7. Ultrahigh Resolution Mass Spectrometry

Water-soluble organic carbon (WSOC) was extracted from quartz fibre filter punches in 5 ml of HPLC grade water using 1.5 h low-speed orbital shaking. The extracts were filtered using 0.2-$\mu$m pore-size PTFE syringe filters (Puradisc 25, Whatman plc, Maidstone, UK) and acidified to pH 2 using 1 M HCl.

30 mg Strata-X cartridges (Phenomenex Inc., Torrance, CA, USA) were used to isolate high molecular weight water-soluble organic carbon from the aqueous extracts. The retained WSOC was eluted twice from the Strata-X cartridges: first, with 1 ml of methanol and water (90/10 v/v); second, with 1 ml of methanol containing 0.3% of aqueous ammonia. Both extracts were combined and the volume was reduced to 1 ml under a gentle stream of nitrogen. The WSOC described in this paper is operationally

defined as the WSOC that is both retained and recovered from the SPE cartridges (SPE-recovered), thus it is not equivalent with the total WSOC. Samples were analyzed using an ultrahigh resolution Orbitrap



Elite mass spectrometer (Thermo Fisher Scientific, Waltham, MA, USA) equipped with a heated electrospray ionization (HESI) source. Mass spectrometer settings for negative HESI included heater temperature of 100ºC, spray voltage of 2.75 V, and capillary temperature of 270ºC. Mass spectra were recorded for m/z 100 to 700 with the resolving power 240,000 at m/z 400. The molecular formulas for

singly charged ions were assigned using an empirical formula calculator (Sierra Analytics Composer64, Modesto, CA, USA). The calculator uses the Kendrick mass defect to sort homologous ion series (species with a given double bond equivalents (DBE) and heteroatom content but differing by increments of -$CH_2$). The calculator was set to allow up to 200 carbon, 1000 hydrogen, 30 oxygen, 3 nitrogen, and 1 sulfur atoms per elemental composition. Additional method details were described in

Mazzoleni et al. (2010) and Putman et al. (2012).

## 3. Results and Discussion

There are two common problems associated with estimation of light absorbing properties of BrC. First, in BB emissions, OC containing BrC is generally mixed with BC, except for pure smoldering

combustion such as encountered for combustion of peat fuels that emits nearly exclusively OC. Therefore, it is often challenging to distinguish between light absorption by BC and by BrC, especially if the BC concentrations are high (e.g., for flaming combustion). Second, most online instruments use 1-7 fixed wavelengths for light absorption measurements (Drinovec et al., 2015, 2017; Lewis et al., 2008; Virkkula et al., 2005), which may not be sufficient to fully characterize spectral absorption properties of

BB BrC emissions (Andreae and Gelencsér, 2006).

In contrast to online instruments, the approach used in this study – measurements of BrC light absorption in extracts – eliminates the interference of BC absorption and allows recording high-resolution (i.e., ~ 1 nm) absorption spectra over a large wavelength range (i.e., 190 – 900 nm, for this work).

### 3.1 Comparison of Absorbance



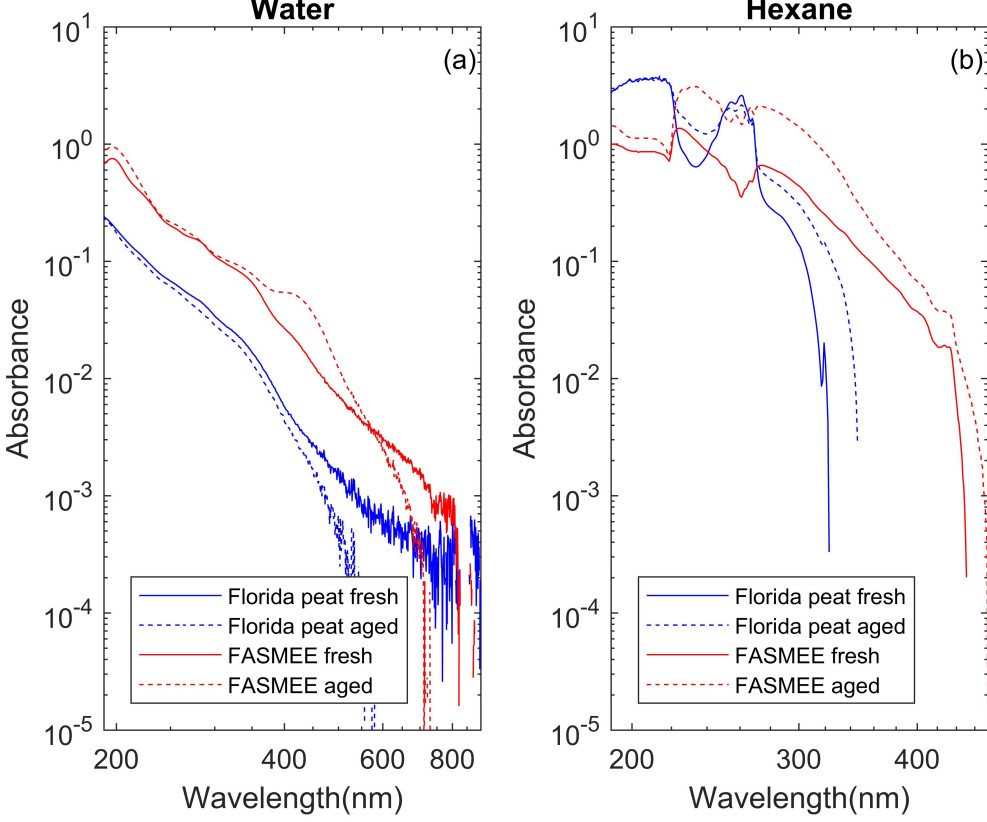

**Figure 2**. Light absorbance spectra of water (2a) and hexane (2b) extracts from representative samples of biomass-burning emissions from two representative fuels. Solid and dashed lines correspond to fresh and aged BB emissions, respectively.

5    Figure 2 shows absorbance spectra measured for water (Fig. 2a) and hexane (Fig. 2b) extracts of fresh and aged BB emissions of two fuels (Florida Peat, FASMEE). Absorbance data measured with the spectrophotometer were plotted on a log-log scale for easier visualization of AAE (eq. 6), directly displaying AAE as negative slope and curvature representing deviations from the power law.

Absorbance of fresh and aged emissions varied between fuels and the spectrophotometer signal was
10  noisy above a wavelength of 600 nm. For peat samples (e.g., Florida peat), absorbance of both fresh and




aged water extracts decreased steadily towards longer wavelengths but we did not observe such steady decrease in the hexane extract, instead fluctuations were observed even in the shorter wavelengths. The absorbance of aged BBOA is less than fresh BBOA for water extracts but for the aged BBOA hexane extract, the absorbance was higher than that of the fresh hexane extract close to 250 nm. In the case of

5 the fuels that undergo flaming combustions (e.g., FASMEE), the absorbance from aged BBOA water extracts were lower than that of the fresh BBOA water extracts at longer wavelengths (>600); in addition, an increase in the absorbance was observed in the 450 – 500 nm wavelength range. Absorbance values for aged BBBOA hexane extract of FASMEE fuel were higher than fresh BBOA in all wavelengths. The increase of absorbance for FASMEE aged BBOA can be attributed to the type of

10 combustion (See Supplement 1). For better quantitative comparison, total solar weighted absorbance and MAC values are compared in section 3.2, 3.4 and 3.6.

### 3.2. Absorbance of Polar and Non-Polar Fractions

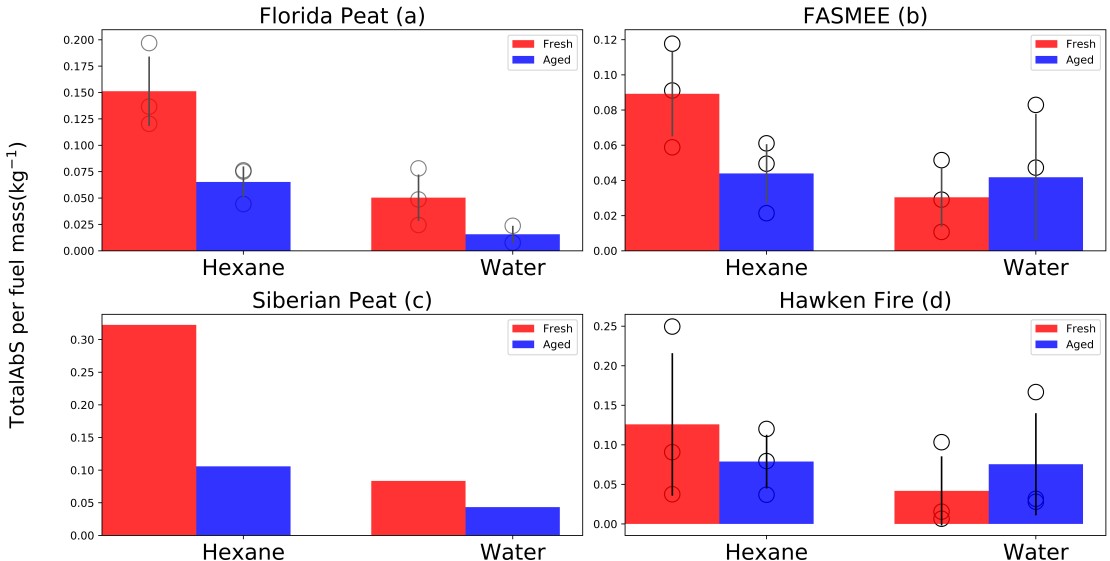

**Figure 3.** Comparison of total absorbance (eq. 3) per fuel mass consumed for fresh and aged aerosols in

water and hexane extracts, normalized to 1-mL extraction volumes. Bars, circles, and error bars



represent the mean values, individual observations, and standard deviations, respectively. No error bars are given for Siberian peat, because only one experiment was performed with this fuel.

To compare the contribution of polar and non-polar fractions to the light absorbance, the following steps were taken. Wavelength-dependent absorbance of each extract was recalculated using the Beer-
Lambert law to that of a solution that would have been obtained if the collected aerosol was extracted in 1 ml of solvent. This was necessary to account for variations in solvent volumes used for aerosol extraction (Section 2.4). Fig. 3 shows the normalized absorbance for polar and non-polar extracts of fresh and aged BB aerosols. While most previous studies have focused on light absorption by the water-soluble fraction of BB aerosols (Hecobian et al., 2010; Xie et al., 2017; Zhang et al., 2013b), very little
attention was given to the light absorption by the water-insoluble part of BB aerosols(Chen and Bond, 2010) Our results suggest that for fresh aerosols from all fuels except for Siberian peat, non-polar (hexane) extracts absorb more than polar (water) extracts (Fig. 3). For fresh aerosols, the ratios of total absorbance in the hexane extract to the water extract are 3.00 (Florida peat), 2.92 (FASMEE), and 3.01 (Hawken Fire). This ~3-fold higher light absorption of the non-polar fraction (hexane extract)
necessitates a detailed chemical speciation analysis to yield further insight into the light absorbing compounds that can be found in non-polar fraction.

Figure 3 shows the absorbance of fresh samples with red bars and aged samples with blue bars. Aging in the OFR changed light absorption properties for BB aerosol extracts for all four fuels, even though there is a significant variability among replicate measurements (shown as points in Fig. 3). For Florida
peat combustion, the total absorbance per consumed fuel mass of aerosol hexane extracts decreased from 0.15 $kg^{-1}$ for fresh aerosol to 0.065 $kg^{-1}$ for aged aerosol. Absorbance of Florida peat combustion water extracts decreased by an even larger factor due to aging (from 0.05 to 0.015 $kg^{-1}$). A similar trend was observed for Siberian peat combustion extracts, where absorbance in the hexane extracts decreased due to aging from 0.058 to 0.032 $kg^{-1}$, while for the water extracts its decrease was from 0.084 to 0.043
$kg^{-1}$. In contrast, light absorbance in water extracts of aged BBOA from FASMEE and Hawken Fire fuels increased due to aging from 0.03 to 0.042 $kg^{-1}$ and from 0.042 to 0.079 $kg^{-1}$, respectively. No increase in absorbance due to aging was observed for hexane extracts of FASMEE and Hawken Fire fuel combustion aerosols, where absorbance decreased from 0.089 to 0.044 $kg^{-1}$ (FASMEE) and from




0.13 to 0.079 kg$^{-1}$ (Hawken Fire).

To quantify the statistical significance of the effect of aging on absorbance values, we computed pairwise t-statistic for fresh and aged aerosol extracts. The p-values from the pairwise t-test for hexane extracts showed statistical significance for Florida peat ($p$=0.04), and FASMEE fuel ($p$=0.016). No
statistically significant difference was observed in the case of the Hawken Fire ($p$=0.442). For Siberian peat we could not perform a statistical significance test since no replicates were collected. The p-values computed for all water extracts showed no statistical difference in absorbance between fresh and aged aerosol extracts (p>0.16). Given the high variability in the observed absorbance values among individual burns, it is possible that a larger number of experiments would have better constrained the
effect of aging on absorbance. However, given the resources available for this study, it was not possible to collect more replicate samples. Moreover, the p-values are higher for the emissions from the flaming fuels (e.g., 0.442 for Hawken Fire fuel combustion emissions in hexane extract) compared to the smoldering fuels (e.g., 0.04 for Florida peat combustion emissions in hexane extract). These higher p-values associated with the flaming fuels can be attributed to the more chaotic nature of flaming
combustion.

The difference in the effect of aging on light absorption by polar extracts between peat and forest fuels is remarkable. While aging decreased the total absorbance of the polar fraction for peat fuels, the opposite was observed for forest fuels. The increase in light absorption of forest fuels is mainly due to the increase in absorption in the 380-500 nm range (Fig. 2). This suggests the formation of new organic
substances that absorb light in this range. Flaming fuels typically had emissions with higher NO$_x$ concentration than smoldering fuels (Supplementary Fig. 2). The presence of NO$_x$ may lead to the formation of nitrogen-containing aromatic compounds during aerosol aging (Kahnt et al., 2013; Kitanovski et al., 2012). These compounds can absorb in the 450 - 550 nm range (Liu et al., 2016b) and therefore, can cause an absorbance increase in this spectral range. High NO$_x$ emissions observed during
the combustion of FASMEE and Hawken fire fuels may be explained by two phenomena: elemental composition of fuel and high temperature burning conditions. Both the FASMEE and Hawken fire fuels had leaves and branches, which may result in greater nitrogen content compared to peats where



most of the nitrogenous compounds are bacterially decomposed during peat formation (Reddy, K. Ramesh; DeLaune, 2008) leaving only carbon and oxygen in the refractory fraction of peat.

### 3.3. Effect of NO$_x$ on Aged Aerosol

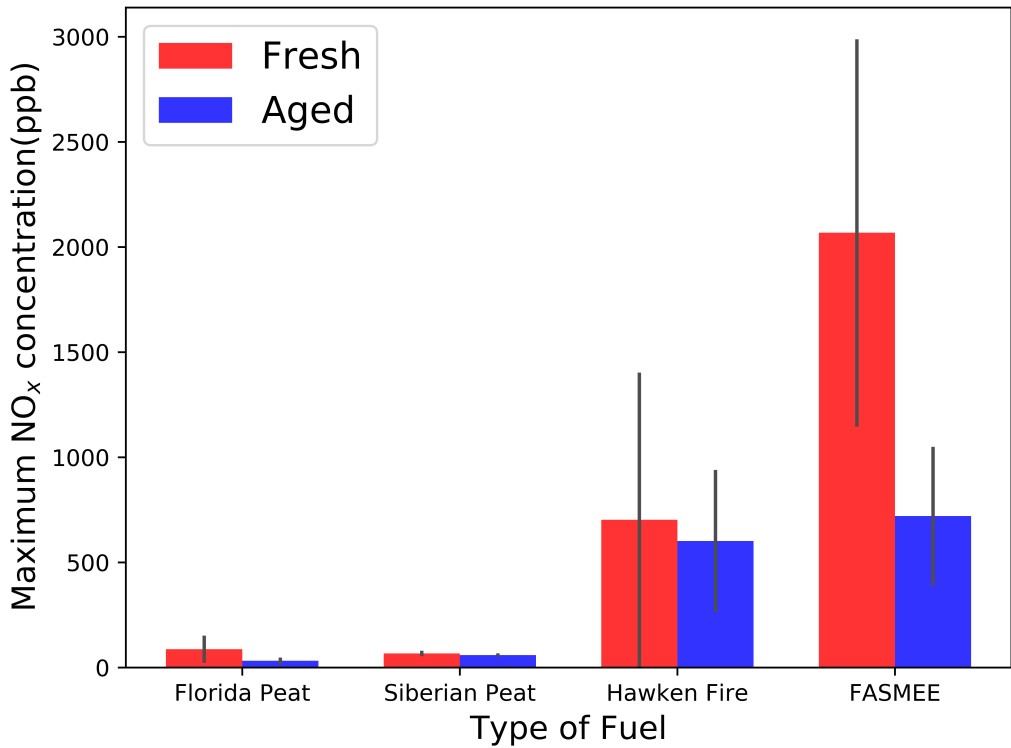

**Figure 4.** Maximum NO$_x$ concentration for emissions from the combustion of different fuels. Average values plotted from replicates as bar and error bars are standard deviations of the mean from three replicate burnings for each fuel.

10  NO$_x$ concentrations reach their maximum in the beginning of each burn (Supplementary Material, Fig. S2). Figure 4 shows maximum NO$_x$ concentrations for the combustion of each fuel. In the case of flaming fuels (Hawken Fire and FASMEE), the maximum NO$_x$ concentrations were up to 800 ppb and





2000 ppb, respectively. In the case of peat fuels (Florida and Siberian), $NO_x$ concentrations did not exceed 100 ppb. This difference is probably due to a much faster fuel consumption rate in higher temperature during flaming combustion (~5.33 g min$^{-1}$) than smoldering combustion of peats (~0.83 g min$^{-1}$) (Supplementary Material, Fig. S1).

We suspect that high $NO_x$ concentrations may affect formation of nitrogen-containing Secondary Organic Aerosols (SOAs) (Laskin et al., 2009; Lin et al., 2015, 2017). The effect of $NO_x$ on the formation of SOA has been highly debated and not sufficiently explored to date. However, experimental chamber studies with representative precursors (Kleindienst et al., 2004; Liu et al., 2016) and model estimates based on ambient measurements (Henze et al., 2007) provide compelling evidence in favor of

increasing SOA yields in the presence of $NO_x$. The mechanism for SOA formation varies with organic precursor type and the amount of $NO_x$ present (Ng et al., 2007a, 2007b, 2008). Nitro-organic compounds, especially the nitro-aromatic compounds, are abundant in the atmosphere and can be used as a tracer for biomass-burning emissions (Iinuma et al., 2010). Previous studies on the evolution of SOA in the presence of $NO_x$ was mostly focused on total SOA yields, without investigating light

absorption properties and compositional variability. Recently, (Liu et al., 2016b) estimated the effect of photochemical aging in the presence of $NO_x$ on light absorption properties of aerosols produced from different primary OA precursors. They reported a large increase in the MAC values in the 300 to 500 nm wavelength range for SOA derived from aromatic precursors in a $NO_x$-rich environment compared to $NO_x$-free conditions. Interestingly, there was no distinct change in the MAC values for SOA derived

from aliphatic precursors under high $NO_x$ conditions. We observed an increase in absorbance values in the range of 380 to 500 nm for the FAASME and Hawken fire aerosol extracts (Fig. 1), which is in agreement with Liu et al. (2016) results. This suggests that the primary precursors for secondary emissions from the combustion of FASMEE and Hawken Fire emissions are mostly aromatic in nature.



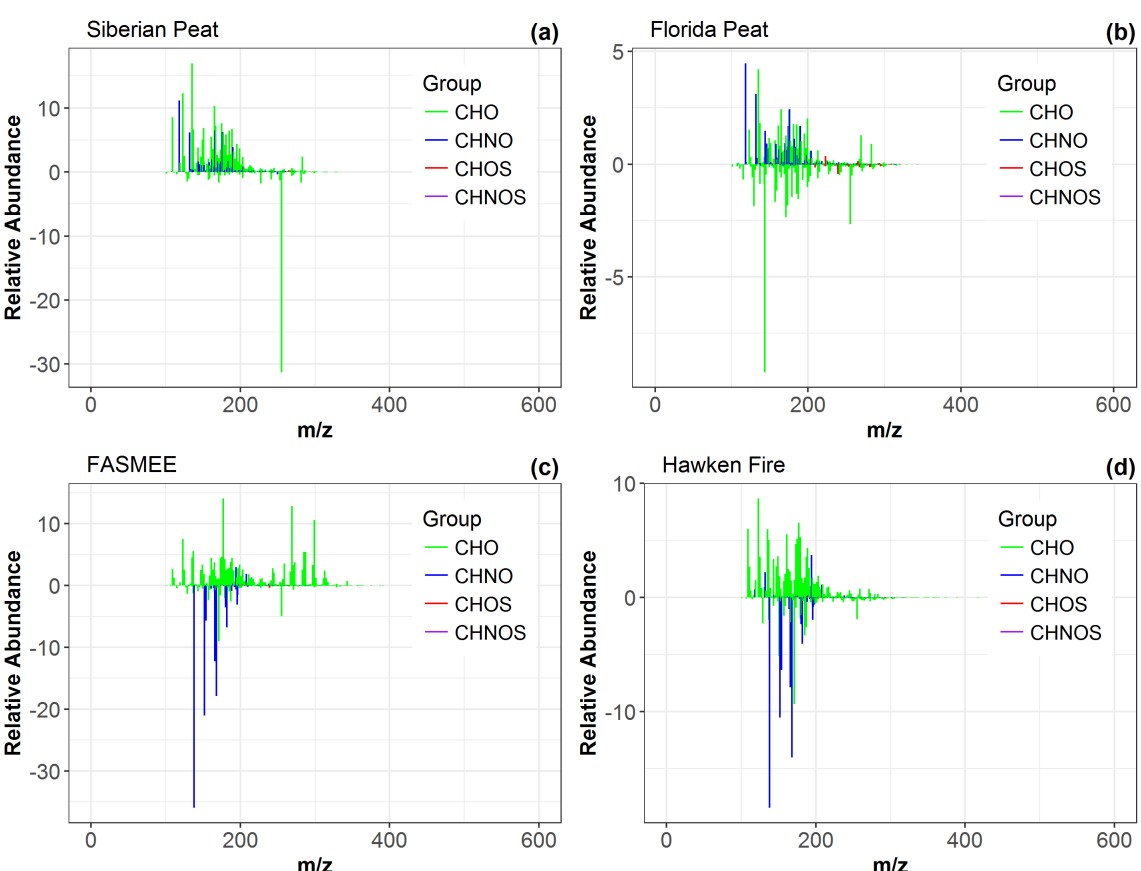

**Figure 5.** Ultrahigh resolution mass spectrometry difference mass spectra for the emissions from the combustion of the representative fuels (smoldering (peat) fuels (on top, (a) & (b)) and flaming fuels (on
5  bottom, (c) & (d))

In order to confirm the presence of newly formed nitrogen-containing SOA during OFR oxidation, the high molecular weight fraction of aqueous extracts of fresh and aged BB samples were analyzed using ultrahigh resolution Orbitrap Elite mass spectrometry. We assigned CHO, CHNO, CHOS, and CHNOS



molecular formulas to the ions detected in the electrospray negative ion mode. Figure 5 shows the difference mass spectra of the assigned molecular formulas for the fresh and aged BB emissions of the analyzed fuels. For each fuel sample, the normalized relative abundances of the aged emissions were subtracted from the normalized relative abundances of fresh BB emissions. Thus, the signals with

positive relative abundances represent the prominent molecular formulas of fresh BB emissions, while signals with negative relative abundances represent the prominent molecular formulas for the aged emissions. No prominent CHNO molecular formulas were detected in the OFR aged aerosol from the combustion of Siberian and Florida peat fuels (Fig. 5a, b). However, a large number of CHNO species were detected in the OFR aged aerosol from the combustion of the FASMEE and Hawken fire fuels

(Fig. 5c, d). This result suggests that CHNO organic compounds were formed during the aging/oxidation process of flaming combustion emissions from FASMEE and Hawken fire fuels. These observations are consistent with the hypothesis that organic nitrogen compound formation occurs during the oxidative aging of flaming combustion fuels when the concentration of nitrogen oxides was high (Fig. 5). DBEs can increase with aging because non-polar aromatic species may become partially

oxidized facilitating their ionization in the aged samples. Although CHNO species were formed in the peat fire combustion experiments, in contrast to the flaming fuel experiments, we did not observe high relative abundances of them nor a large number of uniquely formed species in the aged extracts.



### 3.4. Light Absorption by HUmic LIke Substances (HULIS)

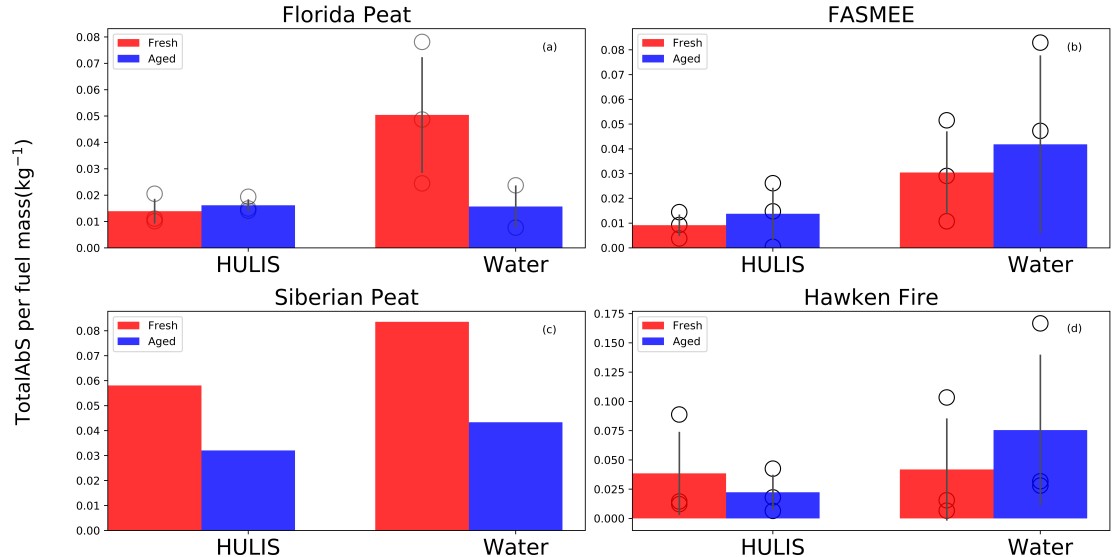

**Figure 6.** Comparison of total absorbance per fuel mass consumed for fresh and aged aerosols in water
and HULIS extracts, normalized to 1-ml extraction volumes. Bars, circles, and error bars represent the
mean values, individual observations, and standard deviations, respectively. No error bars are given for
Siberian peat, because only one experiment was performed with this fuel.

It has been reported that high molecular weight organic species (or HULIS) comprise a substantial
fraction of water-soluble organic aerosols (Graber and Rudich, 2006). HULIS can originate in the
atmosphere either from primary emissions (Lin et al., 2010) or via oligomerization of primary
precursors during SOA formation (Samburova et al., 2005). HULIS can play a significant role in
hygroscopic growth and cloud condensation nuclei activity of ambient aerosols (Dinar et al., 2006;
Gysel et al., 2004). In addition, solar light absorption by HULIS is of special interest with respect to its
impact on direct aerosol radiative forcing and climate change (Hoffer et al., 2006). We isolated the
HULIS fraction from water extracts using the SPE approach. In this study, HULIS fractions isolated
from water extracts of fresh and aged BB emissions were characterized with UV-Vis spectrophotometry




and compared with the spectra of the total water-soluble extracts (Fig. 6). The total solar weighted absorbance per fuel mass consumed for HULIS (e.g., for Florida peat (fresh): 0.14 kg$^{-1}$) is mostly lower compared to the total absorbance per fuel mass consumed for water extracts (e.g., for Florida peat (fresh): 0.5 kg$^{-1}$), except for Hawken fire fuel.

Hawken fire fuel was mostly comprised of cheatgrass and shrubs, characteristic of the semi-arid lower slopes of the Carson Range of the Sierra Nevada, USA. For the fresh Hawken fire BBOA, the total absorbance per fuel mass consumed of the HULIS fraction isolated from the water extract was 0.038 kg$^{-1}$ where that of water extract was 0.041kg$^{-1}$ (Fig. 6). This shows that light absorption by water soluble compounds emitted from the Hawken fire was dominated (~92%) by high molecular weight organic

species (HULIS). We also noticed that the total solar weighted absorbance for water extract of fresh Hawken fire BBOA (0.041 kg$^{-1}$) (Fig. 6d) increased by ~83% (0.075 kg$^{-1}$) (Fig. 6d) after aging. At the same time, a distinct decrease in the absorbance of the HULIS fraction (~42%) was observed upon aging (from 0.038 kg$^{-1}$ to 0.022 kg$^{-1}$). Although the points are variable ($p>0.4$), the decreased absorbance of the HULIS fraction indicates that there may be a greater contribution from lower

molecular weight species after oxidation of the Hawken fire BBOA.

Both fresh and aged BBOA from the FASMEE fuel show the HULIS fraction absorbance is 30 to 40 % of the total water-soluble fraction of aerosol (Fig. 6b). For Siberian peat, the HULIS fraction absorbance was 50 to 55% of the total water-soluble fraction (Fig. 6c), for both the fresh and aged BBOA. The large fraction of HULIS absorption compared to the total water soluble BBOA from all fuels

necessitates detailed chemical characterization. For Florida peat, the total absorbance of the HULIS fraction increased by only ~16% (from 0.139 kg$^{-1}$ to 0.161 kg$^{-1}$) and this was not statistically significant (p=0.676). For the FASMEE fuel, the increase in the total absorbance of "fresh" HULIS (0.091 kg$^{-1}$) after OFR oxidation (0.138 kg$^{-1}$) was also not statistically significant (p=0.536).

**3.5 Effect of pH**



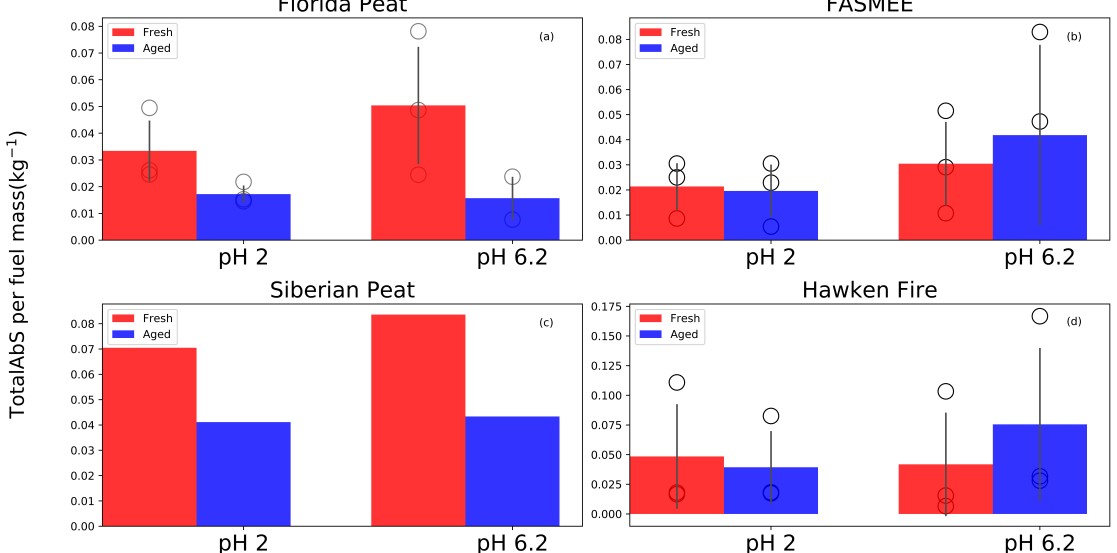

**Figure 7.** Comparison of total absorbance per fuel mass consumed for the fresh and aged aerosols in water (pH = 6.2) and water extracts (pH = 2), normalized to 1-ml extraction volumes. Bars, circles, and error bars represent the mean values, individual observations, and standard deviations, respectively. No
error bars are given for Siberian peat, because only one experiment was performed with this fuel.

Light absorption by water-soluble ambient BrC was shown to depend on pH (Phillips et al., 2017). In our experiments, the pH of biomass-burning water extracts ranged from 6.2 to 6.4. However,
atmospheric aerosol pH often ranges from 0.5 to 3 (Bougiatioti et al., 2016; Weber et al., 2016). Atmospheric processing and mixing with acidic aerosols could decrease the pH of BB aerosols and thus affect their light absorption properties. To investigate the effect of pH on light absorption for our BBOA water extracts, we performed a series of experiments in which these extracts were acidified to pH = 2.

As pH was decreased from ~6.3 to 2, a decrease in absorbance was observed for BBOA water extracts
from all fuels, and this decrease was more distinguishable towards longer wavelengths (> 500 nm) (see Supplementary Material, Fig. S4). It was reported in the literature (Teich et al., 2017; Zhao et al., 2015)





that BB aerosols contain organic compounds with aromatic functional groups. At lower pH, protonation of functional groups occurs and can be the potential reason for the decrease in absorbance. A similar absorption enhancement in the ultraviolet and visible region of the spectrum was reported for soil humic acid (Tsutsuki and Kuwatsuka, 1979) and fulvic acid (Baes and Bloom, 1990) extracts.

In the present study, we found that for the fresh BBOA, the total absorbance (per fuel mass) decreases with the decrease in pH for all fuels (e.g., Florida peat (fresh): from 0.05 kg$^{-1}$ to 0.033 kg$^{-1}$), except for the Hawken fire fresh BBOA, where a small increase the in total absorbance values (from 0.042 kg$^{-1}$ to 0.049 kg$^{-1}$) was observed (Fig. 7). However, the p values (0.213 for Florida peat, 0.270 for FASMEE, 0.094 for Hawken fire) from a pairwise t-test do not permit any definitive conclusions about the effect

of pH on light absorption. The total absorbance values also decreased (e.g., FASMEE: from 0.042 kg$^{-1}$ to 0.019 kg$^{-1}$) for all of the aged BBOA, but these changes were also not statistically significant (p>0.3).

As was discussed above, we have observed the increase in absorbance between fresh and aged neutral water extracts (pH = 6.2-6.4) for flaming fuels (e.g., Hawken fire: from 0.042 kg$^{-1}$ to 0.076 kg$^{-1}$, p-value = 0.158). However, for the acidified water extracts (pH=2) of BBOA from flaming fuels, a decrease in

the absorbance between fresh and aged BBOA was observed (e.g., Hawken fire: from 0.049 kg$^{-1}$ to 0.039 kg$^{-1}$), though this change was not statistically significant (p =0.444).





### 3.6. Light Absorption Efficiency of Polar and Non-Polar Extracts

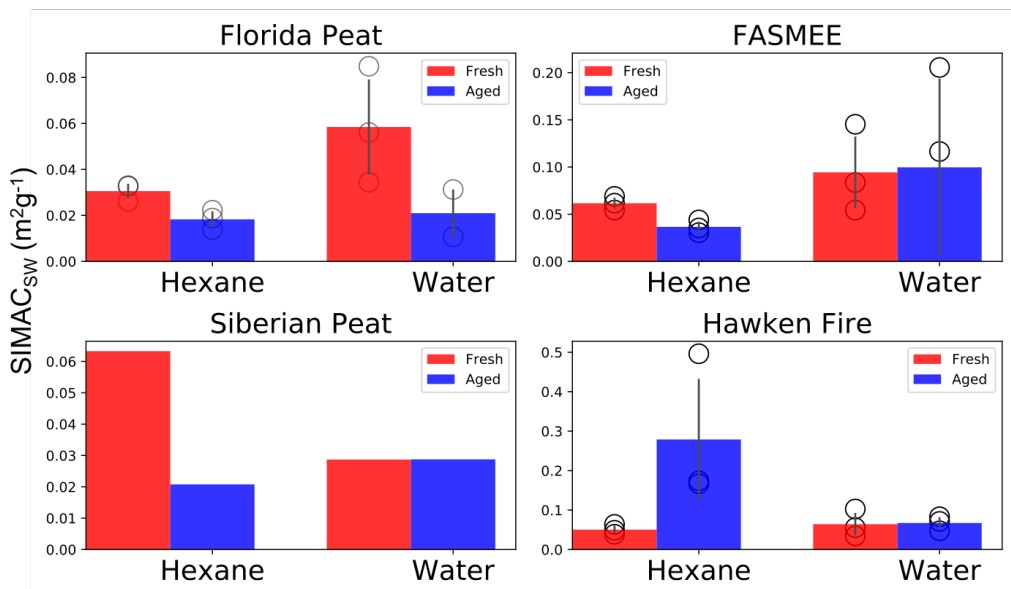

**Figure 8.** Comparison of the total solar-weighted spectrally integrated mass absorption coefficient ($SIMAC_{sw}$) for the fresh and aged aerosols in water and hexane extracts, normalized to 1-ml extraction volumes. Bars, circles, and error bars represent the mean values, individual observations, and standard deviations, respectively. No error bars are given for Siberian peat, because only one experiment was performed with this fuel.

Figure 8 shows the $SIMAC_{sw}$ values of hexane and water extracts of fresh and aged aerosols produced by combustion of the four fuels. $SIMAC_{sw}$ values, which describe mass-based light absorption efficiency, were calculated using the equations provided in section 2.6. In general, $SIMAC_{sw}$ values for all fuels were greater for the water extracts than for the hexane extracts, except for the Siberian peat BBOA.. The highest $SIMAC_{sw}$ values (~2.8 $m^2$ $g^{-1}$) were found for hexane extracts of Hawken fire aged aerosols, whereas the lowest value (~0.02 $m^2$ $g^{-1}$) was observed for hexane extracts of aged Florida aerosols. For example, the $SIMAC_{sw}$ of the Florida peat BBOA hexane extracts was reduced from ~ 0.03 $m^2$ $g^{-1}$ to ~ 0.018 $m^2$ $g^{-1}$ upon oxidation in the OFR, which is almost a 40% reduction. This change





is statistically significant ($p$=0.007). An approximately 65% reduction in the $SIMAC_{sw}$ of the water extracts upon oxidation was observed for the Florida peat BBOA, but the result has a low statistical significance ($p$=0.184). Also, the $SIMAC_{sw}$ values for the hexane extracts of aged Siberian Peat BBOA was about three times lower than that of fresh BBOA, while for the water extracts, the $SIMAC_{sw}$ did not

change significantly (< 4%) upon oxidation.

For the fuels with a dominant flaming combustion phase, the $SIMAC_{sw}$ values of aged BBOA water extracts were higher than those of fresh BBOA water extracts. For example, $SIMAC_{sw}$ values for the aged FASMEE BBOA water extracts were ~5% higher than those for the fresh BBOA extracts. This is mostly due to the increase in the absorbance in the 380 - 550 nm wavelength range for the aged samples

(see Fig. 2), but this increase was not supported by the statistical significance test ($p$=0.911). For hexane extracts with the same FAASME fuel, Figure 8 shows that the $SIMAC_{sw}$ values of aged BBOA particles decreased compared to those of the fresh ones. This change is statistically significant ($p$=0.019). *TotalAbS* values for the FAASME BBOA hexane extracts are also in agreement with the general trend found for $SIMAC_{sw}$ values (See Fig. 3) that decreased ~42% due to aging

In summary, we conclude that the light absorbing BrC compounds present in SOA derived from open flaming combustion of our samples from alpine or boreal forests have higher light absorption efficiencies compared to the compounds emitted from smoldering combustion of peat fuels. Most of the previous research has considered $MAC_{365}$ as a proxy for light absorbing properties of biomass burning aerosols (e.g., Liu et al., 2013; Zhang et al., 2013a). We compared our reported $SIMAC_{sw}$ values with

$MAC_{365}$ values computed from the same experiments (in Table 1). The Fresh/Aged ratios of either $SIMAC_{sw}$ or $MAC_{365}$ are supposed to be larger than one if the absorbance of BBOA decreases after aging and smaller than one when absorbance of BBOA increases upon aging. We observed an underestimation of the $MAC_{365,}$ while predicting a decrease in light absorption upon oxidation (e.g., Florida peat BBOA water extract: Fresh/Aged ratio with $MAC_{365}$ is 2.09 where $SIMAC_{sw}$ ratio is 6.04).

Also, an overestimation of $MAC_{365}$ was found for the Hawken fire fuel BBOA, while predicting the increase in light absorption after aging ($MAC_{365}$ =0.159 where $SIMAC_{sw}$ = 0.956 for Hawken fire)





**Table 1**

| Fuels | Water | | Hexane | |
|---|---|---|---|---|
| | SIMAC$_{sw}$ Ratio | MAC$_{365}$Ratio | SIMAC$_{sw}$ Ratio | MAC$_{365}$ Ratio |
| **Florida Peat** | 6.04 | 2.09 | 1.67 | 1.38 |
| **FAASME** | 0.95 | 1.04 | 1.68 | 1.69 |
| **Hawken fire** | 0.96 | 0.16 | 0.18 | 0.18 |
| **Siberian peat** | 1.15 | 0.81 | 3.63 | 3.59 |

5  **3.7. Change in Absorption Ångström Exponent (AAE) with Aging/Oxidation**

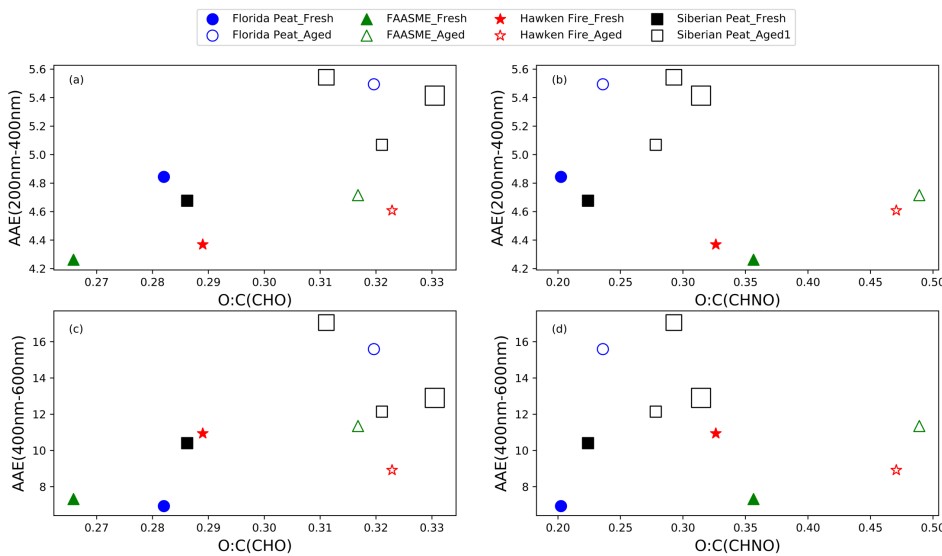

10  **Figure 9.** Absorption Ångström Exponent (AAE) for the water extracts as a function of oxidation. Solid symbols represent fresh aerosols for a fuel type and open symbols of the same shape represent the aged aerosols for the same fuel. For the Siberian peat, the size of the symbol increases with the oxidation voltage in the OFR reactor.



AAE is a key parameter characterizing the spectral dependence of aerosol and bulk (e.g., extracts) light absorption (Moosmüller et al., 2011). BC aerosol from fossil fuel burning typically has AAE values close to one, which signifies a wavelength-independent refractive index (Lack et al., 2014). Aerosol

AAE values slightly greater than one indicate the presence of BC coatings or of some BrC mixed with BC (Gyawali et al., 2009; Pokhrel et al., 2016). High AAE values have been reported for OC emitted by mostly smoldering combustion. Examples include bulk AAE values of up to ~7 for HULIS extracts (Hoffer et al., 2006) and aerosol AAE values of up to ~10 for aerosols from peat combustion (Chakrabarty et al., 2016).

The molecular O:C ratio is an important indicator of oxidation in aerosol samples (Aiken et al., 2008). The relative intensity weighted O:C ratios, calculated from the negative ion ultrahigh resolution mass spectra of SPE-recovered WSOC ranged from 0.27 to 0.33 for the CHO molecular formulas and 0.20 to 0.48 for the CHNO molecular formulas. The bulk AAE values for the water extracts have been calculated (see section 2.5) and are plotted in Fig. 9 as a function of the O:C ratios. AAE determined by

linear regression from 200 to 400 nm [AAE $_{(200-400)}$] and from 400 to 600 nm [AAE $_{(400-600)}$] are shown in panels a, b and c, d, respectively, as function of the O:C ratios for the CHO (on left) and the CHNO (on right) molecular formulas. The solid symbols correspond to fresh emissions from combustion of a particular fuel and the void symbols represent the aged samples for the same fuels. For Siberian peat, the size of the symbols increases with the oxidation voltage applied in the OFR reactor.

The AAE $_{(200-400)}$ values ranged from 4-5.5 and increased with oxidation. For all aged emissions, the AAE $_{(200-400)}$ values were larger than those for fresh emissions indicating a stronger wavelength dependence of absorption in the 200 – 400 nm wavelength range for aged emissions. We observed a similar trend for AAE $_{(400-600)}$ with a single exception for the emissions from the Hawken fire, where the AAE $_{(400-600)}$ values decreased after aging. However, the AAE $_{(400-600)}$ values were much higher (i.e., 7 -

17) than the AAE $_{(200-400)}$ values and their changes during aging/oxidation were also more pronounced (e.g., for Florida peat smoke from 6.93 to 15.59). In conclusion, the aging of BB emissions generally (but not always) increases the AAE values for the water extracts and these AAE values can be unusually large in the 400 to 600 nm wavelength range; this is mostly due to the enhanced absorption around 450





nm as can be seen in Fig. 2. Overall, the aged BBOA from both flaming (e.g., FASMEE fuel) and smoldering (e.g., Florida peat fuels) combustion, showed an increase in the peak relative intensity weighted O:C ratios with a larger increase for the flaming combustion experiments (e.g., from 0.33 to 0.47 for Hawken fire fuel). In conclusion, it can be inferred that more oxygenated compounds were

5 found in CHNO molecular formulas from aged samples. This supports our hypothesis regarding organic nitrogen compound formation from the flaming emissions. To investigate the nature of the BBOA, we analyzed the ultrahigh resolution MS data for our samples. The elemental ratios of O:C and H:C are used to visualize the molecular nature of BBOA (Van Krevelen, 1950). The elemental ratios do not give exact structures but they provide insight regarding the extent of oxidation and saturation. The

10 comparison of fresh to aged FASMEE fuel aerosol indicates that species uniquely observed in the aged FASMEE fuel are more oxidized than the unique species of the fresh FASMEE fuel (Fig. 10).



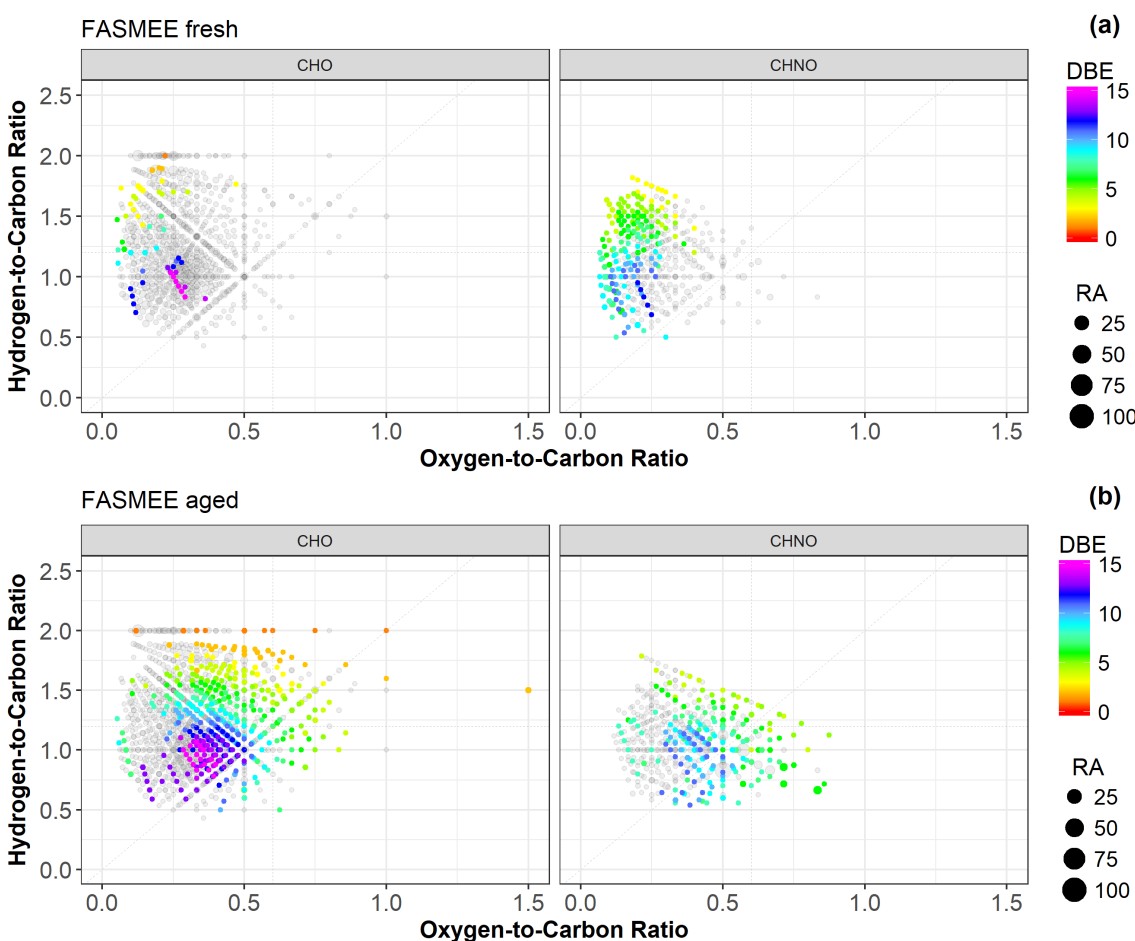

**Figure 10.** Van Krevelen plots of H:C and O:C ratios for the negative ion ultrahigh resolution mass spectra of fresh (a) and aged (b) FASMEE water-soluble aerosol. CHO and CHNO molecular formulas

5 are shown separately. Grey circles indicate common molecular formulas observed in both the fresh and aged samples, colored circles indicate the unique molecular formulas for the fresh or aged water-soluble



aerosol. The color represents the number of double bond equivalents (DBE) and the size of the circles indicates the normalized relative abundance of the peaks in the mass spectra.

Figure 10 shows a loss/transformation of some of the water-soluble organic ions with O:C ratio < 0.3 observed in fresh water-soluble aerosol (colored circles in Fig. 10a) and a formation of mostly more oxidized, with O:C > 0.25 compounds (colored circles in Fig. 10b) during the aging of FASMEE aerosol. Several uniquely observed CHO molecular formulas appear to have formed during the oxidation of FASMEE fuel emissions. Double bond equivalents (DBE) are used for the characterisation of the degree of unsaturation in a molecular formula and represents the sum of rings and double bonds. Both CHO and CHNO molecular formulas formed during the aging process, including a large number of compounds with a high degree of unsaturation (DBE 10-15). The highly unsaturated compounds may contribute to the light absorption of the aged FASMEE aerosol water extract. Aiken et al., (2008) performed high resolution TOF AMS analysis on aerosols collected from laboratory combustion of lodgepole pine and sagebrush seperately. The O:C ratios of their laboratory experiments were 0.3 and 0.4, which are lower than that of ambient samples (> 0.5) collected in northern New Mexico during wildfires. Ambient samples, supposedly photochemically aged and thus were more oxidized which was reflected in the increase of the O:C ratio. In our study, we observed a similar increase in the oxidation for the PAM-aged water-soluble aerosol collected from combustion experiments of the FASMEE fuel.



### 3.8. Imaginary Part of the Bulk Refractive Index of Water Extracts

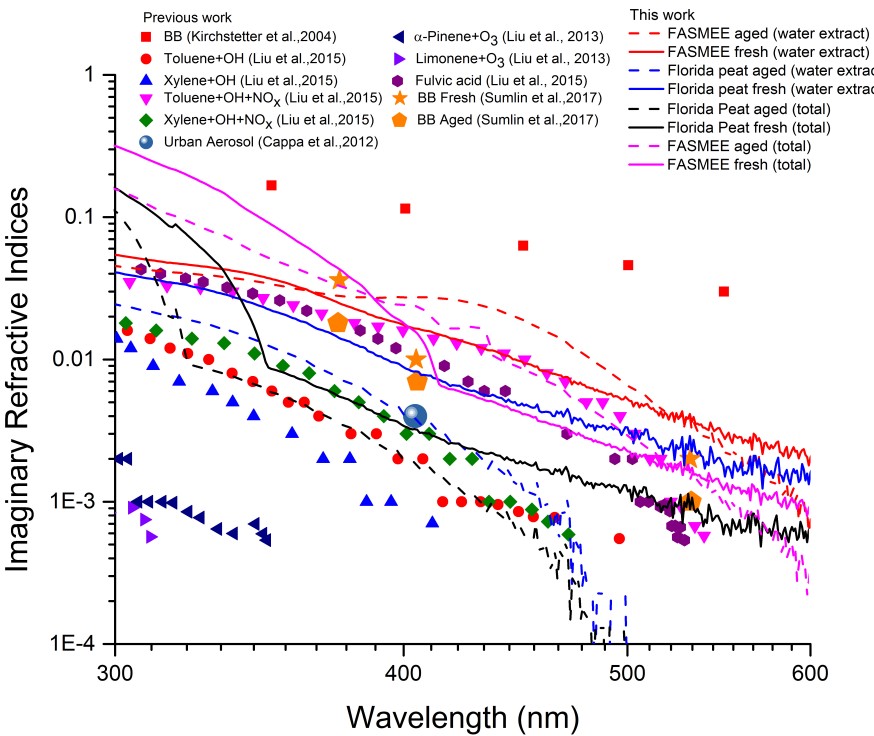

**Figure 11.** Comparison of the imaginary part of the bulk refractive indices from previous work with water extracts and total (hexane + water) extracts of the fresh BBOA (solid lines) and aged BBOA (dotted lines) as a function of wavelength.

Figure 11 shows a comparison of the bulk imaginary refractive indices retrieved for water extracts and the total (hexane + water) extract of BBOA measured in this study with the imaginary refractive indexes reported in the literature. The previously reported data include those from early work by Kirchstetter et al., (2004) on BB aerosols, biogenic SOA (Liu et al., 2013b), anthropogenic SOA (Liu et al., 2015)





generated from laboratory precursor oxidation experiments, urban BrC (Cappa et al., 2013) and Mie theory retrieved imaginary refractive indices from a similar campaign conducted at Washington University of St. Louis (Sumlin et al., 2017). The imaginary part of the bulk refractive indices was retrieved separately for water (Fig. (S5)) and hexane extracts (Fig. (S6)) and then simple mass mixing rules were used to compute the total contribution (Eq. 11 a, b, c). In Fig. 11, the bulk refractive indices for both the water and the total were plotted for comparison. Bulk refractive indices for all fuels (both fresh and aged) are higher at shorter wavelengths and decrease gradually toward longer wavelengths, which is typical for BrC aerosols. The total bulk imaginary refractive indices were higher than water extract only at shorter wavelengths, because the hexane (non-polar) fraction absorbs mostly in the blue and the near-UV part of the spectrum. For fuels that undergo smoldering combustion (e.g., Florida peat), the bulk imaginary refractive indices were smaller for aged BBOA compared to fresh BBOA. On the contrary, a flaming fuel, FASMEE, has very different properties, with the imaginary refractive index of the aged BBOA being smaller than that of fresh BBOA below 400 nm and above 550 nm, while between 400 and 550 nm the imaginary refractive index of aged BBOA was higher than that of the fresh BBOA as indicated by the obvious "hump" in Fig. 11. An enhancement of the imaginary refractive indices was also overserved by Liu et al., (2015) when an anthropogenic precursor was oxidized in the presence of $NO_x$ to generate SOA. This is consistent with our hypothesis regarding the role of $NO_x$ in the absorption enhancement for flaming fuels.

**Summary and conclusions**

In this study, we analysed the absorption properties of biomass-burning aerosols from four different globally and regionally important fuels that represent both smoldering and flaming combustion processes. Our goal was to understand how the light absorbing properties of these combustion experiments was affected by aging (OFR aging) and what compounds (polar or non-polar) contribute to the light absorption of BB aerosols. We found that the non-polar fraction absorbs more than the polar fraction. However, most of the light absorption by the non-polar fraction was attributed to UV and the shorter part of the visible wavelength range. Light absorption by the HULIS fraction was also compared with the light absorption properties of the total water extracts and we found that for the Hawken fire





fuel emissions, the light absorbing compounds were most likely high molecular weight organic species. As the pH of the freshly emitted BBOA can decrease due to atmospheric processing and mixing with more acidic inorganic aerosols, we also compared the light absorption properties of acidified and non-acidified water extracts. We found that the $TotalAbS$ decreased with the decrease in pH, which can be

explained by potential protonation of functional groups at lower pH (pH=2). Light absorbing properties do change upon aging/oxidation and we were able to distinguish the change in the light absorption characteristics of flaming and smoldering combustion emissions. For flaming fuel combustion samples, the absorbance values decreased for non-polar hexane extracts upon the oxidation/aging but in the case of the polar extracts an increase in the absorbance values was observed, especially in the 450-500 nm

wavelength range. We also observed significantly higher NOx levels during flaming fuel combustion (e.g., $[NO_x]_{max}$ = 2068 ppb) comparing to smoldering combustion (e.g., $[NO_x]_{max}$ =88 ppb for Florida peat). Results from the ultrahigh resolution mass spectrometry analysis showed unique organic-nitrogen species that likely formed during aerosol aging/oxidation in the presence of $NO_x$. However, the reason behind this high $NO_x$ emission is still unknown and whether $NO_x$ emission is related to the fuel nitrogen

content or the burning conditions is a matter of future investigation.

**Acknowledgements.**

This research was supported by the Nation Science foundation (NSF) under Grant numbers AGS-1544425, AGS-1408241, NASA ROSES under Grant number NNX15AI48G, and internal funding from the Desert Research Institute. The authors would like to thank Dr. Anna Tsibar (Moscow State

Lomonosov University, Moscow, Russia) for providing Siberian peat fuel, Anna Cunningham (DRI) for technical assistance with extraction procedure, and Lan Gao for assisting in data analysis.



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
