# Peer review of "Light absorption by polar and non-polar aerosol compounds from laboratory biomass combustion"

_Atmospheric Chemistry and Physics, 2018_

## Referee Comment (RC1) · Anonymous Referee #3 · 11 Apr 2018

also see attached file

Review of ACP-2018-161

This study examined the optical and chemical properties of fresh and aged aerosols from laboratory biomass burning (BB). The effect of polarity, pH, molecular weight, biomass fuel type, and combustion conditions [flaming versus smoldering] on the aerosol optical properties were reported. The aged aerosols were produced in an oxidation flow reactor (OFR). The optical and chemical properties of the aerosols were measured using UV-Visible spectroscopy and high-resolution LC-MS, respectively. A spectrally integrated mass absorption efficiency was calculated, and Van Krevelen diagrams were used to visualize chemical information about the extent of oxidation. The study found that non-polar matter in the organic aerosols was more absorbing than the

polar fraction; although, water soluble matter in the aged aerosol was more absorbing during flaming. Low pH aerosol was less absorbing than at neutral pH. The study also provided convincing evidence that the NOx formed during flaming combustion yields organic nitrogen compounds that are strongly absorbing at shorter wavelengths.

This is a useful, well-done study that is within the ACP scope. Biomass burning is an important source of organic aerosol in the atmosphere, and its optical and chemical properties are key to understanding the effects of aerosols on climate, air quality, and human health. The results presented are novel in that aerosol optical properties are measured after being fractionated in-bulk using multiple chemical methods. The investigation of fresh versus aged aerosol also produced an interesting contrast. The examination of peat is something that is in its infancy and more peat burning studies are needed owing to the increasing potential for peat mega-fires forming globally. Mostly, the chemical and optical data synched together nicely and evidence-based conclusions are given. Although, with so many variables, sometimes it was difficult to track the rationale underlying an experiment and the obtained results. The comments below are meant to produce further clarification regarding the experimental results. The paper may also benefit from a forward-looking statement that describes specifically where some of these results may be useful or applied. Such a statement can come in the summary or at the end of the introduction. After these minor revisions, the paper merits publication.

Additional comments:

1. P1, line 10: It sounds like BrC is the only light-absorbing component in BB aerosols. This opening sentence should be re-phrased for clarity sakes. 2. P5, line 23: Please describe the pre-firing technique (i.e., add a temp. and time). 3. P6, line 12: "and" should be or. 4. P7, lines 10 and on: It would be a good idea to tell us how much time it took to complete each type of burn and provide a better idea of the sample time weighting. 5. P7, lines 21-22: It's not clear why the NO2/m was handled this way. Is the assumption that the fuel burned the same over the entire test even though

sampling time was limited? 6. P.9, lines 1-15: Where is the extraction efficiency calculation? It would be useful to see what fraction of aerosol was removed from the filter using each solvent? 7. P10, lines 6-8: This sentence needs to be clearer. 8. P11, lines 1-3: Same thing, this sentence requires clarification. 9. General comment: It would be beneficial if there was some additional discussion about what is expected to happen during the aging in the OFR. Was the OFR producing new particles? Or was it changing the existing particles via heterogenous chemistry? Or both? Which of these processes/mechanisms is likely to dominate? What are the particle loss characteristics of this specific OFR? With the knowledge that BB fuel types produced different aerosol types, is one aerosol type more susceptible to loss than another? More information that helps us assess these results should be added here. 10. P17, line 1-2: Revise this sentence for clarity. 11. P17, lines 13-15: Not 100% convinced about this. It may simply be that one is measuring less of something thus the uncertainty is a larger fraction of the measurement. 12. P17, lines 26-27: The fuels could have been tested for their N content. Are those data available? 13. Figure 5: Mention in the caption that this set of plots is for WSOC. 14. P21, lines 14-15: Regarding DBEs, it sounds like that bias is built into the measurement via the ionization chamber. Should that be further addressed? 15. P26, lines 11-13: Not sure Fig. 8 supports this observation in general for both aged and fresh aerosol. Please clarify. 16. P27, lines 15-18: Does it mean anything that smoldering biomass emits more per unit mass of fuel burned? We're getting deep into the manuscript now, and it may be good to remind the reader that everything is normalized to weight or volume or…..... If it is? 17. Figure 10: Why was no attempt made to examine the hexane soluble extracts? May want to briefly raise this point in the Experimental section or while or after discussing the virtues of this result. 18. P25, lines 4-18: This discussion starts earlier at page 21 (lines 14-15), but doesn't really take off until here. 19. Figure 11: This is a nice summarizing figure. Although, why are only select experiments from the current study shown here? Consider showing all the experiments.

Please also note the supplement to this comment:
https://www.atmos-chem-phys-discuss.net/acp-2018-161/acp-2018-161-RC1-supplement.pdf

―――――――――――――――――――――――

---

## Referee Comment (RC2) · Anonymous Referee #2 · 19 Apr 2018

This manuscript describes the study of the absorption spectra of water- and hexane-soluble components of biomass burning aerosols. Aerosols were generated from selected fuels under both flaming and smoldering conditions and these were studied nascently and after chemical aging in an oxidation flow reactor. The authors found that the solar-weighted hexane-soluble fraction was 2-3 times more absorbing (per mass of fuel consumed) than the water-soluble fraction. They also found that absorbance of the hexane-soluble fraction of all samples decreased after aging while the aged water-soluble fractions of two of the samples increased. Futhermore, the shape of the spectrum, as indicated by the absorption Ångström exponent, was found to change upon aging. Higher NOx levels were observed from the flaming combustion, and the high-resolution mass spectra show evidence for organic-nitrogen species that may be

formed during aging in the presence of NOx.

Overall, this manuscript provides an interesting comparison between polar and non-polar light-absorbing components of model biomass burning aerosols. The data and results are presented clearly, though some of the interpretations are confusing or not consistent with the figures. For example, it is stated that the hexane-soluble extracts were more absorbing than the water-soluble extracts for all samples except for the Siberian peat; yet, Figure 3 clearly shows that this is true for all four samples, including the Siberian peat. Such an inconsistency makes it confusing for the reader and under-mines his/her confidence in the interpretation of the data, especially since this is one of the primary conclusions from the study.

Given that relatively few studies have explored the differences in optical properties of polar and non-polar aerosol extracts, this work serves to advance the understanding of brown carbon in biomass burning aerosols. However, despite the plethora of data presented, including high-resolution mass spectra and pH dependence of the absorption measurements, there are very few insights or general conclusions drawn from this work. Additionally, the absorbance data are presented as solar-weighted, i.e. weighted by the actinic solar spectrum, which makes interpretation difficult. For example, Figure 3 shows that the (solar-weighted) hexane-soluble fraction has a larger absorbance than the (solar-weighted) water-soluble fraction for all four fuels studied, but is this true for all wavelengths, or is the difference larger at some (e.g. UV) wavelengths than at others? While Figure 2 does compare the spectra from the two fractions, it is not clear how the spectral dependencies differ. Perhaps a plot of the ratio of the two spectra would be helpful.

In summary, the work presented here might be worthy of publication in ACP but only if the conclusions drawn were more clearly stated. As it stands now, it is a jumble of results riddled with typos and mistakes making it difficult for the reader to appreciate what has been learned from these experiments. The conclusions need to be gener-alized to provide some more fundamental insight into the nature of the differences in

absorption between the polar and non-polar fractions.

Specific comments:

1. Page 1, line 24: It should be clearly stated that the non-polar fraction is "2-3 times more absorbing" when weighted by the solar spectrum and normalized to mass of fuel consumed. 2. Page 1, lines 25-26: It is stated that "an increased absorbance was observed for water extracts of oxidized/aged emissions," but that appears to be true only for two of the four samples, FASMEE and Hawken Fire; the other two, Florida Peat and Siberian Peak, show a decrease upon aging (Figure 3). 3. Page 1, lines 27-28: The statement "Comparing the absorption Ångström Exponent (AAE) values, we observed changes in the light absorption properties of BB aerosols with aging that was dependent on the fuel types." is vague. What is meant by "light absorption properties"? The only observed difference was in the AAE values themselves. 4. Page 3, line 11: "Just a decade ago …" is followed by a citation to a paper from 2001, 17 years ago. That's not a decade. 5. Page 4, line 22: "muck" is not a very scientifically specific word and should be replaced. 6. Page 10, line 7: "700 to 900 nm and" should read "700 to 900 nm". 7. Page 10, line 8: The use of the term "AbS$\lambda$" is confusing; why not use "Abs$\lambda$"? Why use a capital S? 8. Page 11, line 2: remove "was calculated". 9. Page 16, line 11: The results don't "suggest", they show/indicate/demonstrate. There is no inference in this statement. 10. Page 16, line 11-12: Figure 3 clearly shows that there is more absorbance in the hexane-soluble fraction than the water-soluble fraction for all four samples, including the Siberian peat. 11. Page 16, lines 12-14: It should be made clear here that the "total absorbance" referred to is the solar-weighted total absorbance. 12. Page 19, lines 20-23: First, the authors refer to Fig. 1, but the absorption spectra appear in Fig. 2. Second, "FAASME" should be "FASMEE." Third, the FASMEE and Florida Peat absorption spectra are shown in Fig. 2, not FASMEE and Hawken Fire samples as stated. Fourth, the increase in absorption observed occurred over a wider range of wavelengths than just 380-500 nm. Fifth, are the authors referring to an increase in absorbance for the water-soluble fraction or the hexane-soluble fraction?

Sixth, an increase in absorbance over a wide range of wavelengths such as observed here most certainly does not suggest that the "primary precursors for secondary emissions . . . are mostly aromatic in nature"; this statement is highly speculative and should be removed. 13. Page 24, lines 14-16: how does the decrease in absorbance with decreasing pH compare to the results of (Phillips et al., 2017) mentioned? Is it similar or not, and if not why not? 14. Page 25, lines 1-2: how does the protonation of functional groups explain the decrease in absorption with lower pH? This statement needs to be clarified.

---

## Author Comment (AC1) · 20 Jun 2018

RESPONSES TO REVIEWER 1

This study examined the optical and chemical properties of fresh and aged aerosols from laboratory biomass burning (BB). The effect of polarity, pH, molecular weight, biomass fuel type, and combustion conditions [flaming versus smoldering] on the aerosol optical properties were reported. The aged aerosols were produced in an oxidation flow reactor (OFR). The optical and chemical properties of the aerosols were measured using UV-Visible spectroscopy and high-resolution LC-MS, respectively. A spectrally integrated mass absorption efficiency was calculated, and Van Krevelen diagrams were used to visualize chemical information about the extent of oxidation. The

study found that non-polar matter in the organic aerosols was more absorbing than the polar fraction; although, water soluble matter in the aged aerosol was more absorbing during flaming. Low pH aerosol was less absorbing than at neutral pH. The study also provided convincing evidence that the NOx formed during flaming combustion yields organic nitrogen compounds that are strongly absorbing at shorter wavelengths. This is a useful, well-done study that is within the ACP scope. Biomass burning is an important source of organic aerosol in the atmosphere, and its optical and chemical properties are key to understanding the effects of aerosols on climate, air quality, and human health. The results presented are novel in that aerosol optical properties are measured after being fractionated in-bulk using multiple chemical methods. The investigation of fresh versus aged aerosol also produced an interesting contrast. The examination of peat is something that is in its infancy and more peat burning studies are needed owing to the increasing potential for peat mega-fires forming globally. Mostly, the chemical and optical data synched together nicely and evidence-based conclusions are given. Although, with so many variables, sometimes it was difficult to track the rationale underlying an experiment and the obtained results. The comments below are meant to produce further clarification regarding the experimental results. The paper may also benefit from a forward-looking statement that describes specifically, where some of these results may be useful or applied. Such a statement can come in the summary or at the end of the introduction. After these minor revisions, the paper merits publication.

AC: We thank the reviewer for his/her comments and recommendations. We address comments individually below. Approximate line numbers corresponding to the edited (with markup) manuscript is provided.

Comments: 1. P1, line 10: It sounds like BrC is the only light-absorbing component in BB aerosols. This opening sentence should be re-phrased for clarity sakes.

AC: The sentence was rephrased.

2. P5, line 23: Please describe the pre-firing technique (i.e., add a temp. and time). AC:

The "pre-firing" conditions (temperature and time) for 47-mm quartz filters are added to the text.

3. P6, line 12: "and" should be or.

AC: "and" was changed to "or"

4. P7, lines 13-14: It would be a good idea to tell us how much time it took to complete each type of burn and provide a better idea of the sample time weighting.

AC: Thank you for the recommendation. The following sentence was added:"The duration of smoldering combustion was 58ïĆś27 min on average, whereas the average duration for flaming combustion was 22ïĆś8 min."

5. P7, lines 22-25: It's not clear why the NO2/m was handled this way. Is the assumption that the fuel burned the same over the entire test even though sampling time was limited?

AC: We agree. NO2/m was not used in the present work (only NO2 max values were presented, in ppb). These sentences were removed from the text.

6. P.8, Section 2.4: Where is the extraction efficiency calculation? It would be useful to see what fraction of aerosol was removed from the filter using each solvent?

AC: We thank the reviewer for bringing up this important point. We also feel a calculation of extraction efficiency/percent recovery would be useful as we are comparing absorbance values from water and hexane extracts. Sonication is a standard technique for extraction of water soluble aerosol components from filters for which almost 100% efficiency is assumed (Fang et al., 2015; Gao et al., 2017; Kiss et al., 2002; Samburova et al., 2005). In our study, sonication was performed twice to ensure maximum recovery. An accelerated solvent extractor (ASE) was used to recover hexane soluble fraction from the filter. The ASE uses high solvent pressure and temperature to ensure maximum recovery. A statement to this effect was added to the experimental section (section 2.4, Page 8, Line 25).

7. P10, lines 7-11: This sentence needs to be clearer. AC: The paragraph was revised.

8. P11, lines 3-9: Same thing, this sentence requires clarification. AC: The paragraph was re-written.

9. General comment: It would be beneficial if there was some additional discussion about what is expected to happen during the aging in the OFR. Was the OFR producing new particles? Or was it changing the existing particles via heterogeneous chemistry? Or both? Which of these processes/mechanisms is likely to dominate? What are the particle loss characteristics of this specific OFR? With the knowledge that BB fuel types produced different aerosol types, is one aerosol type more susceptible to loss than another? More information that helps us assess these results should be added here.

AC: We appreciate reviewer's concern about more detailed description about OFR characterization and fate of organics during aging. Detailed OFR characterization and discussion of various mechanism occurring in the OFR and their timescales can be found in another manuscript associated with this work (Bhattarai et al., 2018) (under review). We have added a sentence in section 2.3, page 7, lines 10-12.

To address reviewer's questions we will briefly summarize the main outcomes of Bhattarai et al., (2018) paper. We found that a wall loss of low-volatile organic compounds (LVOC) and semi-volatile organic compounds (SVOC) is negligible (0.1%) in the OFR. However, about 25% of primary organic aerosol mass is lost due to fragmentation reactions. An approximately equal amount of secondary products condenses onto primary particles. We did observe new particle formation in the OFR.

Particle loss in the accumulation size range was found to be less than 10%. Since the aerosol mass was concentrated in this size range, we do not expect such losses to be significant for our absorbance measurements which are mass-based. Diffusional particle losses depend only on particle size and not on chemical properties. Thus, we do not expect any differences in particle losses among different aerosols tested in our

study.

10. P18, lines 3-4: Revise this sentence for clarity. AC: The sentence was rephrased.

11. P18, lines 1-2: Not 100% convinced about this. It may simply be that one is measuring less of something thus the uncertainty is a larger fraction of the measurement. AC: We agree with the reviewer and the sentence is removed from the text.

12. P18, lines 14-17: The fuels could have been tested for their N content. Are those data available? AC: Thank you for a valuable suggestion. We plan to measure the nitrogen content of the fuels in future. Unfortunately, these data are in not available at the present time.

13. Figure 5: Mention in the caption that this set of plots is for WSOC. AC: We thank the reviewer for pointing this out. All ultrahigh resolution mass spectrometry results demonstrated in this work are from water soluble organic carbon (WSOC). We have added WSOC in the Figure 5 caption as per suggestion.

14. P22, lines 13-15: Regarding DBEs, it sounds like that bias is built into the measurement via the ionization chamber. Should that be further addressed?

AC: This is a well-known bias with electrospray ionization. In the negative ion mode, only the molecules with an active H (e.g., carboxyl groups) can be deprotonated. In this case, we wanted to see which new species were formed by oxidation, so the bias is quite useful. To clarify this, the following sentence was added to page 13 line 11.

15. P27, lines 11-12: Not sure Fig. 8 supports this observation in general for both aged and fresh aerosols. Please clarify.

AC: The statement on page 27 holds true only for fresh aerosols. This point is clarified in the revised manuscript as per reviewer's suggestion

16. P28, lines 17-18: Does it mean anything that smoldering biomass emits more per unit mass of fuel burned? We're getting deep into the manuscript now, and it may be

good to remind the reader that everything is normalized to weight or volume or. . . .... If it is?

AC: In page 28, line 17-18 Our results showed that the secondary organic aerosols (SOA) produced by OFR oxidation (OFR-aged) of BB aerosols from flaming combustion type fuel have larger light absorption efficiencies. In contrast, the OFR-aged BB emission generated by smoldering combustion of peat fuels absorb less than fresh BB emissions from the same fuels. The absorption efficiency was normalized by mass of OC. The units' explanation was added (Figure 8 capture and P. 28, line 17)

17. Figure 10: Why was no attempt made to examine the hexane soluble extracts? May want to briefly raise this point in the Experimental section or while or after discussing the virtues of this result.

AC: Oxidized (ionized) molecules were not expected to be observed in the hexane extracts. Our intention was to inform the water-soluble OM composition since it's the most likely fraction to contain the oxidized components that are amenable to ESI. In future studies, we can use APPI (Atmospheric Pressure Photoionization) to further explore the composition, however, that was deemed beyond the scope of the current project.

18. P25, lines 4-18: This discussion starts earlier at page 21 (lines 14-15), but doesn't really take off until here. AC: We apologize, but we are confused regarding this comment and are not sure what the reviewer is referring to.

19. Figure 11: This is a nice summarizing figure. Although, why are only select experiments from the current study shown here? Consider showing all the experiments.

AC: Thank you! We also think that imaginary refractive indices for all fuel extracts (polar and non-polar) are important to present with the data reported from the previous works. However, when we tried to include all data sets for all fuels, the figure became too busy and hard to read. For this reason, we selected only two fuels and refractive

indices for water and total extracts for the comparison. In the Supplementary Material, we presented the imaginary refractive indices of water (Fig. S5) and hexane (Fig. S6) extracts from all fuels and for both fresh and aged aerosols.

Please also note the supplement to this comment:
https://www.atmos-chem-phys-discuss.net/acp-2018-161/acp-2018-161-AC1-supplement.pdf
* * *
[Figure]

**Supplement:**

[revised manuscript text omitted]

---

## Author Comment (AC2) · 20 Jun 2018

This manuscript describes the study of the absorption spectra of water- and hexane soluble components of biomass burning aerosols. Aerosols were generated from selected fuels under both flaming and smoldering conditions and these were studied nascently and after chemical aging in an oxidation flow reactor. The authors found that the solar-weighted hexane-soluble fraction was 2-3 times more absorbing (per mass of fuel consumed) than the water-soluble fraction. They also found that absorbance

of the hexane-soluble fraction of all samples decreased after aging while the aged water-soluble fractions of two of the samples increased. Furthermore, the shape of the spectrum, as indicated by the absorption Ångström exponent, was found to change upon aging. Higher NOx levels were observed from the flaming combustion, and the high-resolution mass spectra show evidence for organic-nitrogen species that may be formed during aging in the presence of NOx. Overall, this manuscript provides an interesting comparison between polar and nonpolar light-absorbing components of model biomass burning aerosols. The data and results are presented clearly, though some of the interpretations are confusing or not consistent with the figures. For example, it is stated that the hexane-soluble extracts were more absorbing than the water-soluble extracts for all samples except for the Siberian peat; yet, Figure 3 clearly shows that this is true for all four samples, including the Siberian peat. Such an inconsistency makes it confusing for the reader and undermines his/her confidence in the interpretation of the data, especially since this is one of the primary conclusions from the study. Given that relatively few studies have explored the differences in optical properties of polar and non-polar aerosol extracts, this work serves to advance the understanding of brown carbon in biomass burning aerosols. However, despite the plethora of data presented, including high-resolution mass spectra and pH dependence of the absorption measurements, there are very few insights or general conclusions drawn from this work. Additionally, the absorbance data are presented as solar-weighted, i.e. weighted by the actinic solar spectrum, which makes interpretation difficult. For example, Figure 3 shows that the (solar-weighted) hexane-soluble fraction has a larger absorbance than the (solar-weighted) water-soluble fraction for all four fuels studied, but is this true for all wavelengths, or is the difference larger at some (e.g. UV) wavelengths than at others? While Figure 2 does compare the spectra from the two fractions, it is not clear how the spectral dependencies differ. Perhaps a plot of the ratio of the two spectra would be helpful. In summary, the work presented here might be worthy of publication in ACP but only if the conclusions drawn were more clearly stated. As it stands now, it is a jumble of results riddled with typos and mistakes making it difficult for the

reader to appreciate what has been learned from these experiments. The conclusions need to be generalized to provide some more fundamental insight into the nature of the differences in absorption between the polar and non-polar fractions.

AC: We thank the reviewer for a constructive review of the manuscript. We agree that some conclusions are not clear. We revised our manuscript accordingly. Regarding the use of solar-weighed values, we chose this representation to simplify the comparison between different experiments. Spectral dependence of measured absorbances and how it differs from experiment to experiment can be observed in Fig.11, Fig. S5, and Fig. S6 that show imaginary refractive index values.

Specific comments:

1.Page 1, lines 23-24: It should be clearly stated that the non-polar fraction is "2-3 times more absorbing" when weighted by the solar spectrum and normalized to mass of fuel consumed.

AC: This is an important reviewer's observation. The sentence was re-written "Results of spectrophotometric measurements (absorption weighted by the solar spectrum and normalized to mass of fuel consumed) over the 190 to 900 nm wavelength range showed that the non-polar (hexane-soluble) fraction is 2-3 times more absorbing than the polar (water-soluble) fraction."

2. Page 1, line 26: It is stated that "an increased absorbance was observed for water extracts of oxidized/aged emissions," but that appears to be true only for two of the four samples, FASMEE and Hawken Fire; the other two, Florida Peat and Siberian Peak, show a decrease upon aging (Figure 3).

AC: Yes, the increased absorbance was only for FASMEE and Hawken Fire fuel which comes under flaming combustion type fuel. This condition is added as text in the revised manuscript.

3. Page 1, line 28, P2. Lines 1-3: The statement "Comparing the absorption Ångström
Exponent (AAE) values, we observed changes in the light absorption properties of BB aerosols with aging that was dependent on the fuel types." is vague. What is meant by "light absorption properties"? The only observed difference was in the AAE values themselves.

AC: We agree, "light absorption properties" did not clearly refer to the AAE values. We changed the statement in the revised manuscript.

4. Page 3, line 11: "Just a decade ago . . ." is followed by a citation to a paper from 2001, 17 years ago. That's not a decade.

AC: We agree that "Just a decade ago" is not equivalent to 17 years of time as mentioned by the reviewer. We changed the beginning of the sentence to "Until recently".

5. Page 4, line 22: "muck" is not a very scientifically specific word and should be replaced.

AC: The term 'muck' is replaced with "mucky peat" in the revised version of the manuscript. According to the report of natural resources and conservation service (https://www.nrcs.usda.gov/Internet/FSE_DOCUMENTS/nrcs142p2_053171.pdf), Mucky peat is hemic organic material, which is characterized by decomposition that is intermediate between that of fibric material and that of sapric material

6. Page 10, line 9: "700 to 900 nm and" should read "700 to 900 nm".

AC: The sentence was re-written in the revised manuscript.

7. Page 10, line 11: The use of the term "Abs$\lambda$" is confusing; why not use "Abs$\lambda$"? Why use a capital S?

AC: All capital 'S' was replaced by 's' in Abs$\lambda$ expressions everywhere in the text and Figures.

8. Page 11, line 4: remove "was calculated".

AC: "was calculated" was removed as per suggestion

9. Page 16, line 14: The results don't "suggest", they show/indicate/demonstrate. There is no inference in this statement.

AC: 'suggest' was replaced by 'demonstrate'

10. Page 16, line 14: Figure 3 clearly shows that there is more absorbance in the hexane-soluble fraction than the water-soluble fraction for all four samples, including the Siberian peat

AC: Thank you for pointing this out. The "except for Siberian peat" was mentioned by mistake and has been corrected.

11. Page 16, lines 16: It should be made clear here that the "total absorbance" referred to is the solar-weighted total absorbance.

AC: "total absorbance" was changed on "TotalAbs" in order to maintain proper reference/ consistency to the equations in section 2.6 (Page 10, line 22).

12. Page 20, line 18: First, the authors refer to Fig. 1, but the absorption spectra appear in Fig. 2.

AC: Thank you! In the revised version of the manuscript, we refer to Fig. 2 at this point.

Second, "FAASME" should be "FASMEE."

AC: "FASMEE" was corrected.

Third, the FASMEE and Florida Peat absorption spectra are shown in Fig. 2, not FASMEE and Hawken Fire samples as stated.

AC: The 'typo' was corrected.

Fourth, the increase in absorption observed occurred over a wider range of wavelengths than just 380-500 nm.

AC: Yes, we agree that the increase in absorbance was observed in wider wavelength range than 380-500 nm and hence the range was changed to 380-580 nm in the current manuscript

Fifth, are the authors referring to an increase in absorbance for the water-soluble fraction or the hexane-soluble fraction?

AC: We thank the reviewer for pointing out this mistake. The increase is only observed for water extracts and we corrected that in the manuscript.

Sixth, an increase in absorbance over a wide range of wavelengths such as observed here most certainly does not suggest that the "primary precursors for secondary emissions . . . are mostly aromatic in nature"; this statement is highly speculative and should be removed.

AC: We agree with the reviewer that this statement is highly speculative and it was removed from the text (P.20, Lines 19-20)

13. Page 25, line 8: how does the decrease in absorbance with decreasing pH compare to the results of (Phillips et al., 2017) mentioned? Is it similar or not, and if not why not?

AC: Phillips et al. measured changed in absorbance for a wide range of pH in ambient aerosol samples and they showed that with the decrease of pH, the absorbance also decreased. In the present study, our samples were acidified to pH=2 and we observed a similar trend – absorption decreased. The sentence is added into the text (Page 26, lines 2-4).

14. Page 26, lines 1-2: how does the protonation of functional groups explain the decrease in absorption with lower pH? This statement needs to be clarified.

AC: The organic fraction of BB aerosols is comprised of several chromophores and identification of specific chromophores responsible for light absorbance from different fuels is challenging (Liu et al., 2014). We suspect light absorption by BB aerosols

can be due to the presence nitro-phenols (Mohr et al., 2013), an adduct of amine and carbonyl compounds (Powelson et al., 2014), and charge transfer complexes (Phillips and Smith, 2014). These compound are pH sensitive and lose their ionic structures (responsible for color) on protonation in lower pH (e.g. pH=2).

**Supplement:**

[revised manuscript text omitted]